# GEOMETRY OF NEURAL REINFORCEMENT LEARNING IN CONTINUOUS STATE AND ACTION SPACES

**Saket Tiwari** [*]
Department of Computer Science
Brown University

**Omer Gottesman**
Amazon Web Services

**George Konidaris**
Department of Computer Science
Brown University

## ABSTRACT

Advances in reinforcement learning (RL) have led to its successful application in complex tasks with continuous state and action spaces. Despite these advances in practice, most theoretical work pertains to finite state and action spaces. We propose building a theoretical understanding of continuous state and action spaces by employing a geometric lens to understand the *locally attained* set of states. The set of all parametrised policies learnt through a semi-gradient based approach induces a set of attainable states in RL. We show that the training dynamics of a two-layer neural policy induce a low dimensional manifold of attainable states embedded in the high-dimensional nominal state space trained using an actor-critic algorithm. We prove that, under certain conditions, the dimensionality of this manifold is of the order of the dimensionality of the action space. This is the first result of its kind, linking the geometry of the state space to the dimensionality of the action space. We empirically corroborate this upper bound for four MuJoCo environments and also demonstrate the results in a toy environment with varying dimensionality. We also show the applicability of this theoretical result by introducing a local manifold learning layer to the policy and value function networks to improve the performance in control environments with very high degrees of freedom by changing one layer of the neural network to learn sparse representations.

## 1 INTRODUCTION

The goal of a reinforcement learning (RL) agent is to learn a policy that maximises its expected, time discounted cumulative reward (Sutton & Barto, 1998). Recent advances in RL have lead to agents successfully learning in environments with enormous state spaces, such as games (Mnih et al., 2015; Silver et al., 2016; Wurman et al., 2022), robotic control in simulation (Lillicrap et al., 2016; Schulman et al., 2015; 2017) and real environments (Levine et al., 2016; Zhu et al., 2020; Deisenroth & Rasmussen, 2011; Kaufmann et al., 2023). However, we do not have an understanding of the intrinsic complexity of these seemingly large problems.

We investigate the complexity of RL environments through a geometric lens. We build on the intuition behind the *manifold hypothesis*, which states that most high-dimensional real-world datasets actually lie on or close to low-dimensional manifolds (Tenenbaum, 1997; Carlsson et al., 2007; Fefferman et al., 2013; Bronstein et al., 2021); for example, the set of natural images is a very small, smoothly varying subset of all possible value assignments for the pixels. A promising geometric approach is to model this data as a low-dimensional structure—a *manifold*—embedded in a high-dimensional ambient space. In supervised learning, especially deep learning theory, researchers have shown that the approximation error depends strongly on the dimensionality of the manifold (Shaham et al., 2015; Pai et al., 2019; Chen et al., 2019; Cloninger & Klock, 2020), thus connecting the learning complexity to the complexity of the underlying structure of the dataset. RL researchers have previously applied the manifold hypothesis —- i.e. by hypothesizing that the effective state space lies on a low-dimensional manifold (Smart & Kaelbling, 2002; Mahadevan, 2005; Machado et al., 2017; 2018; Banijamali et al., 2018; Wu et al., 2019; Liu et al., 2021), but the assumption has never been theoretically and empirically validated.

---

[*]Corresponding author: saket_tiwari@brown.edu

RL shares many similarities with control theory (Bertsekas, 2012; 2024). In a control-theoretic framework, the objective is to drive the system, over time, to a desired state or goal. Consequently, theoreticians and practitioners are often interested in the *reachability* of a control system to understand what state is reachable given how system changes under control inputs, i.e. the system dynamics. Locally reachable states are the set of states to which the system can possibly transition, starting from a fixed state, under all *smooth* time variant controls. Control theorists have long studied the set of reachable states (Kalman, 1960) using a differential geometric framework (Sussmann, 1973; 1987). Theoretical research in the study of control systems is often focused on finding necessary and sufficient conditions in the system dynamics such that all states are reachable (Isidori, 1985; Sun et al., 2002; Respondek, 2005; Sun, 2007) under all the admissible time-variant policies. In RL, the objective is to maximize the discounted return via gradient-based updates to the policy parameters, so the focus is on states attained through a sequence of policies determined by these parameters.

Furthermore, a theoretical understanding of states attainable using neural network (NN) policies gives us insight into the geometry and low-dimensional structure of data in RL. This requires utilising an analytically tractable model of NNs. Ever since the remarkable success of neural networks, researchers have developed various theoretical models to better understand their efficacy. A theoretical model intended to study a complex object, such as a neural network, often ends up making simplifying assumptions for tractability. One such theoretical model studies the evolution of neural networks linearly in parameters during training of *wide* neural networks (Lee et al., 2019; Jacot et al., 2018), meaning in a setting where the width approaches infinity. This has helped researchers develop theories of the generalization properties of neural networks (Jacot et al., 2018; Allen-Zhu et al., 2019a; Wei et al., 2019; Adlam & Pennington, 2020). We similarly utilize a single hidden layer neural network model for the policy that is linear in terms of its parameters, not linear in the state, as the width approaches infinity, as has previously been applied to RL (Wang et al., 2019; Cai et al., 2019a).

Within this theoretical framework, we provide a proof of the manifold hypothesis for deterministic continuous state and action RL environments with wide two-layer neural networks. We prove that the effective set of attainable states is subset of a manifold and its dimensionality is upper bounded linearly in terms of the dimensionality of the action space, under appropriate assumptions, independent of the dimensionality of the nominal state space. The primary intuition is that the set of states locally attained are restricted by two factors: **1)** the policy is time invariant and state dependent, and **2)** the set of policies is constrained by the optimization of a *wide*, two-layer neural network using stochastic policy gradients. Our theoretical results are for deterministic environments with continuous states and actions; we empirically corroborate the low-dimensional structure of attainable states on MuJoCo environments (Todorov et al., 2012) by applying the dimensionality estimation algorithm by Facco et al. (2017). To show the applicability and relevance of our theoretical result, we empirically demonstrate that a policy can implicitly learn a low-dimensional representation with marginal computational overhead using the CRATE framework (Yu et al., 2023a;b; Pai et al., 2024). We present an algorithm that does two things simultaneously: **1)** learns a mapping to a local low dimensional representation parameterised by a DNN, and **2)** uses this effectively low-dimensional mapping to learn the policy and value function. Our modified neural network works out of the box with SAC (Haarnoja et al., 2018) and we show significant improvements in high dimensional DM control environments (Tunyasuvunakool et al., 2020).

## 2 Background and Mathematical Preliminaries

We first describe the continuous-time Markov decision process (MDP), which forms the foundation upon which our theoretical result is based. Then we provide mathematical background on various ideas from the theory of manifolds that we employ in our proof.

### 2.1 Continuous-Time Reinforcement Learning

We first analyse continuous-time reinforcement learning in a deterministic *Markov decision process* (MDP) defined by the tuple $\mathcal{M} = (\mathcal{S}, \mathcal{A}, \mathcal{T}, f_r, s_0, \lambda)$ over time $t \in [0, T)$. $\mathcal{S} \subset \mathbb{R}^{d_s}$ is the set of all possible states of the environment. $\mathcal{A} \subset \mathbb{R}^{d_a}$ is the rectangular set of actions available to the agent. $\mathcal{T} : \mathcal{S} \times \mathcal{A} \times \mathbb{R}^+ \to \mathcal{S}$ and $\mathcal{T} \in C^\infty$ is a *smooth* function that determines the state transitions:

$s' = \mathcal{T}(s, a, \tau)$ is the state to which the agent transitions when it takes the action $a$ at state $s$ for the time period $\tau$. Note that $\mathcal{T}(s, a, 0) = s$, which means that the agent's state remains unchanged if an action is applied for a duration of $\tau = 0$. The reward obtained for reaching the state $s$ is $f_r(s)$, determined by the reward function $f_r : \mathcal{S} \to \mathbb{R}$. $s_t$ denotes the state the agent is at time $t$ and $a_t$ is the action it takes at time $t$. $s_0$ is the fixed initial state of the agent at $t = 0$, and the MDP terminates at $t = T$. The agent lacks access to $f$ and $f_r$, and can only observe states and rewards at a given time $t \in [0, T)$. The agent's decision-making process is determined by its policy $\pi : \mathcal{S} \to \mathcal{A}$. Simply put, the agent takes action $\pi(s)$ in state $s$. The agent's goal is to maximize the discounted return $J(\pi) = \int_0^{\mathsf{T}} e^{-\frac{l}{\lambda}} f_r(s_l) dl$, where $s_{t+\epsilon} = \mathcal{T}(s_t, \pi(s_t), \epsilon)$ for small $\epsilon$ and for all $t \in [0, T)$. We define the *action tangent mapping*, $g : \mathcal{S} \times \mathcal{A} \to \mathbb{R}^{d_s}$, for an MDP as

$$\nabla_a f(s, a) = \lim_{\epsilon \to 0^+} \frac{\mathcal{T}(s, a, \epsilon) - s}{\epsilon} = \frac{\partial \mathcal{T}(s, a, \epsilon)}{\partial \epsilon}.$$

Intuitively, this captures the direction of change in the state $s$ after taking an action $a$. We consider the family of control affine systems that represent a wide range of control systems (Isidori, 1985; Murray & Hauser, 1991; Tedrake, 2023), such that $\dot{s}_t = g(s) + \sum_{i=1}^{d_a} h_i(s) a_i$, where $\dot{s}_t$ is the time derivative of the state, $g, h_i : \mathbb{R}^{d_s} \to \mathbb{R}^{d_s}$ are infinitely differentiable (or smooth) functions. Similarly, $\pi(s) = [\pi_1(s), \ldots, \pi_{d_a}(s)]$ is the direction of change in the agent's state following a policy $\pi$ at state $s$ for an infinitesimally short time. The curve in the set of possible states, or the state-trajectory of the agent, is a differential equation whose integral form is:

$$s_t^\pi = s_0 + \int_0^t g(s_l^\pi) + \sum_{i=1}^{d_a} h_i(s_l^\pi) \pi_i(s_l^\pi) dl. \tag{1}$$

This solution is also unique (Wiggins, 1989) for a fixed start state, $s_0$, and Lipschitz continuous policy, $\pi$. The above curve is smooth if the policy is also smooth. Therefore, given an MDP $\mathcal{M}$ and a smooth deterministic policy $\pi \in \Pi$, the agent traverses a continuous time state-trajectory or curve $H_{\mathcal{M}, \pi} : [0, T] \to \mathcal{S}$. The value function at time $t$ for a policy $\pi$ is the cumulative future reward starting at time $t$:

$$v^\pi(s_t) = \int_t^T e^{-\frac{l+t}{\lambda}} f_r(s_l^\pi) dl. \tag{2}$$

Note that the objective function, $J(\pi)$, is the same as $v^\pi(s_0)$. Our specification is very similar to classical control and continuous time RL (Cybenko, 1989; Doya, 2000a) but we define the transitions, $\mathcal{T}$, differently. More recently, researchers have developed the theory for continuous-time RL in a model-free setting with stochastic policies and dynamics (Wang et al., 2020; Jia & Zhou, 2022a).

## 2.2 MANIFOLDS

In practice, MDPs have a low-dimensional underlying structure resulting in fewer degrees of freedom than their nominal dimensionality. In the Cheetah MujoCo environment, where the Cheetah is constrained to a plane, the goal of the RL agent is to learn a policy to make the Cheetah move forward as fast as possible. The actions available to the agent are to provide torques at each of the 6 joints. For example, an RL agent learning from control inputs for the Cheetah MuJoCo environment, one can "minimally" describe the cheetah's state by its "pose", velocity, and position as opposed to the entirety of the input vector. The idea of a low-dimensional manifold embedded in a high-dimensional state space formalises this.

A function $h : X \to Y$, from one open subset $X \subset \mathbb{R}^{l_1}$, to another open subset $Y \subset R^{l_2}$, is a diffeomorphism if $h$ is bijective, and both $h$ and $h^{-1}$ are differentiable. Intuitively, a low dimensional surface embedded in a high dimensional Euclidean space can be parameterised by a differentiable mapping, and if this mapping is bijective we term it a diffeomorphism. Here, $X$ is diffeomorphic to $Y$. A manifold is defined as follows (Guillemin & Pollack, 1974; Boothby, 1986; Robbin et al., 2011).

**Definition 1.** *A subset $M \subset \mathbb{R}^k$ is called a smooth $m$ dimensional submanifold of $\mathbb{R}^k$ iff every point $p \in M$ has an open neighborhood $U \subset \mathbb{R}^k$ such that $U \cap M$ is diffeomorphic to an open subset $O \subset \mathbb{R}^m$. A diffeomorphism, $\phi : U \cap M \to O$ is called a coordinate chart of $M$ and the inverse, $\psi := \phi^{-1} : O \to U \cap M$ is called a smooth parameterisation.*

We illustrate this with an example in Figure 1. Further note that a coordinate chart is called *local* to some point $p \in U \subset M$ the diffeomorphism property holds in the neighborhood $U$. It offers a local "flattening" of the local neighborhood. It is called global if it holds everywhere in $M$ but not all manifolds have a global chart (e.g., Figure 1). If $M \subset \mathbb{R}^k$ is a smooth non-empty $m$-manifold, then $m \leq k$, reflecting the idea that a manifold is of lower or equal dimension than its ambient space. A smooth curve $\gamma : I \to M$ is defined from an interval $I \subset \mathbb{R}$ to the manifold $M$ as a function that is infinitely differentiable for all $t$. The derivative of $\gamma$ at $t$ is denoted as $\dot{\gamma}(t)$. The set of derivatives of the curve at time $t$, $\dot{\gamma}(t)$, for all possible smooth $\gamma$, forms a set that is called the tangent space $T_p(M)$ at the point $p$. For a precise definition, see the appendix A. Taking partial derivatives of $\psi$ with respect to each coordinate $x^j$, we obtain the vectors in $\mathbb{R}^k$:

$$\frac{\partial \psi}{\partial x^j} = \left( \frac{\partial \psi^1}{\partial x^j}, \frac{\partial \psi^2}{\partial x^j}, \ldots, \frac{\partial \psi^k}{\partial x^j} \right)$$

These vectors span the tangent space $T_pM$ at the point $p$. Therefore, locally the manifold can be alternatively represented as the space spanned by the non-linear bases: $\text{Span}(\frac{\partial \psi^1}{\partial x^j}, \frac{\partial \psi^2}{\partial x^j}, \ldots, \frac{\partial \psi^k}{\partial x^j})$.

### 2.3 VECTOR FIELDS, LIE-SERIES, AND CONTROL THEORY

Curves and tangent spaces in manifolds naturally lead to vector fields. In the same way that a curve represents how an agent's state changes continuously, a vector field captures this change locally at every point of the state space. A *tangent vector* can be represented as $X = [v_1, \ldots, v_m]^\mathsf{T}$, where each $v_i$ is a function.

**Definition 2.** *The vector field $X$ is called a $C^r$ vector field if, in any local coordinate chart $(U, \varphi)$ with coordinates $(x_1, \ldots, x_m)$, the components of $X$ in the local basis $\left\{ \frac{\partial}{\partial x_i} \right\}$ are $C^r$ functions. That is, in local coordinates, $X$ can be written as $X(x) = \sum_{i=1}^m v_i(x) \frac{\partial}{\partial x_i}$, where each component function $v_i : U \to \mathbb{R}$ is $C^r$.*

We denote by $V^\infty(M)$ the set of all smooth vector fields on manifold $M$. The rate of change of a function $f \in C^\infty(M)$ at a point $x$ along the vector field $X$ is defined by

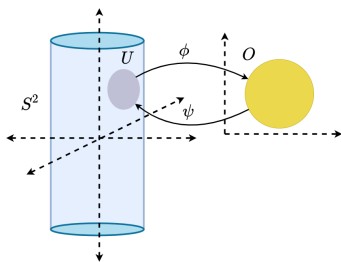

Figure 1: The surface of an open cylinder of unit radius, denoted by $S^2$, in $\mathbb{R}^3$ is a 2D manifold embedded in a 3D space. More formally, $S^2 = \{(x, y, z) | x^2 + y^2 = 1, z \in (-h, h)\}$ where the cylinder's height is $2h$. One can smoothly parameterise $S^2$ as $\psi(\theta, b) = (\sin \theta, \cos \theta, b)$. The coordinate chart is $\phi(x, y, z) = (\sin^{-1} x, z)$.

$$L_X(f) = X(f(x)) = \sum_{i=1}^m v_i \frac{\partial f(x)}{\partial x_i}. \tag{3}$$

Associated with every such vector field $X \in V^\infty(M)$ and $x_0 \in M$ is the integral curve: $x(t)$. Intuitively, following along the direction $X$ for time $t$, the curve starting from $x_0$ reaches the point $x(t)$. The solution to the ODE with the starting condition $x(0) = x_0$ is denoted as the exponential map $e_t^X(x_0)$. One can imagine that vector fields have a connection to policies in the way a policy determines the direction of change. Therefore, it is an effective way to model the change in an agent's state given a vector field and an arbitrary fixed starting state over a time period.

Taylor series help approximate complex functions with polynomials; analogously, we will use the Lie series of the exponential map. To define this expansion, we first recursively define the derivative of Lie $L_X^k(f) = L_X(L_X^{k-1}(f))$ for $k \in \mathbb{N}^+$ where $L_X(\cdot)$ is defined in equation 3. The Lie series of the exponential map with $f(x) = x$ is (Jurdjevic, 1997; Cheng et al., 2011)

$$e_t^X(x) = x + tX(x) + \sum_{l=1}^\infty \frac{t^{l+1}}{(l+1)!} L_X^{l+1}(x). \tag{4}$$

## 3 MODEL FOR LINEARISED WIDE TWO-LAYER NEURAL POLICY

An RL agent in the policy gradient framework (Sutton et al., 1999; Konda & Tsitsiklis, 1999) is equipped with a policy $\pi$ parameterised by parameters $\theta$ and takes gradient ascent steps in the direction of $\nabla_\theta J(\pi(;\theta))$. Suppose that the agent's policy is parameterised by a *wide* two-layer neural network policy. This update direction can be estimated in different ways (Williams, 1992; Kakade, 2001). Such an algorithm generates a sequence of parameters:

$$\theta^{\tau+1} \leftarrow \theta^\tau + \eta \nabla_\theta J(\theta), \tag{5}$$

where $\eta$ is the learning rate. In our setting, a neural RL agent parameterises the policy as a two-layer neural network with smooth activation. We highlight the salient details below.

### 3.1 LINEAR PARAMETERISATION OF NEURAL POLICY

For the set of permissible policies we consider the family of two-layer feed forward neural networks with GeLU activation (Hendrycks & Gimpel, 2016), which is a smooth analog of the popular ReLU activation (Nair & Hinton, 2010). We follow the parameterisation for a two-layer fully connected neural network employed by Cai et al. (2019b) and Wang et al. (2019) for analysis of RL algorithms, which is also used in theoretical analyses of wide neural networks in supervised learning (Allen-Zhu et al., 2019b; Gao et al., 2019; Lee et al., 2019). For a weight vector $W$ of the first layer and weights $C$ for the last layer a shallow, two-layer, fully connected neural network is parameterised as:

$$f(s; W, C) = \frac{1}{\sqrt{n}} \sum_{\kappa=1}^{n} C_\kappa \varphi(W_\kappa \cdot s), \tag{6}$$

where $W \in \mathbb{R}^{nd_s}$ is the vector of first layer parameters where each $W_k$ is a $d_s$ length vector block and therefore the complete vector $W = [W_1, W_2, ..., W_n]$, $\varphi$ is GeLU activation, and $B \in \mathbb{R}^{d_a \times n}$ is a matrix comprised of $n$ column vectors of dimension $d_a$ denoted $C_k$, meaning $B = [C_1, C_2, ..., C_n]$. Here $n$ is the width of the neural network. The parameters are randomly initialised i.i.d as $B_k \sim \text{Unif}(-1, 1)$ and $W_k \sim \text{Normal}(0, I_{d_s}/d_s)$, where $I_{d_s}$ is a $d_s \times d_s$ identity matrix and Unif is the uniform distribution. During training, Cai et al. (2019b) and Wang et al. (2019) only update $W$ while keeping $B$ fixed to its random initialisation despite which, for a slightly different policy gradient based learning, the agent learns a near optimal policy. Researchers study neural networks in simplified theoretical settings to advance the understanding of a complex system while keeping the mathematics tractable (Li & Yuan, 2017; Jacot et al., 2018; Du et al., 2018; Mei et al., 2018a; Allen-Zhu et al., 2019b). While this shallow model of neural networks does away with complexity from multiple layers, it captures the over-parameterization in NNs.

Let $W^0$ be the initial parameters of the policy network defined in Equation 6. A *linear approximation* of the policy is defined as

$$f^{\text{lin}}(s; W) = f(s; W^0) + \nabla_\theta f(s; \theta)|_{\theta=W_0}(W - W^0) = f(s; W^0) + \Phi(s; W_0)(W - W^0) \tag{7}$$

where $\Phi(x; W_0) = \frac{1}{\sqrt{n}}\left[C_1^0 \varphi'(W_1^0 \cdot s)s^\mathsf{T}, C_2^0 \varphi'(W_2^0 \cdot s)s^\mathsf{T}, ..., C_n^0 \varphi'(W_n^0 \cdot s)s^\mathsf{T}\right]$ is a $d_a \times nd_s$ feature matrix for the input $s$, $\varphi'$ is the gradient of GeLU function w.r.t the input, $\cdot$ represents the dot product, and the matrix $\Phi$ formed by the concatenation of $d_a \times d_s$ matrices $B_k \varphi'(W_k^0 \cdot s)s^\mathsf{T}$ for $k = 1, ..., n$. This results in a matrix of size $d_a \times nd_s$. $W$ is an $nd_s$ vector as described above. We will omit the parameters $W, W^0$, and $B$ from the representation of policies when there is no ambiguity. It is a linear approximation because it is linear in the weights $W$ and non-linear, within $\Phi$, in the initial weights $W^0$ and the state $s$. This leads us to the definition of the family of linearised policies for a fixed initialisation $W^0$, similar to Wang et al. (2019).

**Definition 3.** *For a constant $r > 0$, and fixed $W_0$. For all widths $n \in \mathbb{N}_{\geq 0}$, we define*

$$\mathcal{F}_{W_0, r, n} = \left\{ \hat{f} = \frac{1}{\sqrt{n}} \sum_{\kappa=1}^{n} C_\kappa^0 \varphi'(W_\kappa^0 \cdot s)s \cdot W_\kappa : ||W - W^0|| \leq r \right\}.$$

This linearised approximation of the policy simplifies our analysis of the set of reachable states. We further note that it might seem restrictive to consider a network without bias, but we can extend this analysis by adding another input dimension, which is always set to 1.

## 3.2 Continuous Time Policy Gradient

Under the parameterisation described above, the sequence of neural net parameters as described by the updates in equation 5, are determined by the semi-gradient update direction

$$\nabla_\theta J = \mathbb{E}\left[\nabla_a Q^\pi(s, a, t)\nabla_\theta f^{\text{lin}}(s; W)\right],$$

where the expectation is over the visitation measure $\rho_\pi$ and $Q^\pi : \mathcal{S} \times \mathcal{A} \times [0, T]$ is the action-value function that represents the value of taking a constant action $a$ at time $t$. For further details see Appendix H. As is usually done, the following stochastic gradient based update rule approximates the true gradient for the policy parameters

$$W_{(k+1)\eta} - W_{k\eta} = \frac{\eta}{B}\sum_{b=1}^{B}\nabla_a Q^{W_{k\eta}}(s_b, a_b, t_b)\Phi(s_b; W_0), \tag{8}$$

where $W_{k\eta}$ represents the parameters after $k$ gradient steps with learning rate $\eta$, $Q^{W_{k\eta}}$ is the action-value function associated with policy parameterised by $f^{\text{lin}}(; W^{k\eta})$, $\mathbb{B}_{W_{k\eta}} = \{(s_b, a_b, t_b)\}_{b=1}^{B}$ is randomly chosen batch of data from samples of the SDE (Doya, 2000b; Jia & Zhou, 2022a)

$$dS_t = \left(g(S_t) + \sum_{i=1}^{d_a} h_i(S_t)f_i^{\text{lin}}(s; W_{k\eta})\right)dt + \sigma(S_t)dw_t,$$

where $W_0 = W^0$ (see Section 3.1), $w_t$ is the $d_s$ dimensional Wiener process where $\sigma : \mathbb{R}^{d_s} \to \mathbb{R}^{d_s \times d_s}$ is the exploration component of the agent. We assume access to an oracle that gives us the gradients $\nabla_a Q^{W_{k\eta}}$, which do not need to be true in practice. Therefore, a sample $\mathbb{B}_W$ is an i.i.d. set of samples from $\{1, \ldots, N'\}$, for large $N'$, of size $B$. Thus we can write the expectation of the gradient update as follows

$$\mathbb{E}_{\mathbb{B}_W}\left[\frac{\eta}{B}\sum_{b=1}^{B}\nabla_a \hat{Q}^{W_{k\eta}}(s_b, a_b, t_b)\Phi(s_b; W_0)\right] = \frac{\eta}{N'}\sum_{i=1}^{N'}\nabla_a \hat{Q}^{W_{k\eta}}(s_i, a_i, t_i)\Phi(s_i; W_0),$$

where we have an appropriate function $Q$ such that the above condition is satisfied. Let the term on the right hand side be denoted by $\nabla_W J(W)$ in the limit $N' \to \infty$. Here, $\sigma$ is the exploration component of the dynamics. We re-write the update rule from equation 8 as follows,

$$W_{(k+1)\eta} - W_{k\eta} = \eta\nabla_W J(W)|_{W=W_{k\eta}} + \eta\xi(W_{k\eta}, \mathbb{B}_{W_{k\eta}}) = G(W_{k\eta}, \eta), \tag{9}$$

where $\xi(W_{k\eta}, \mathbb{B}_{W_{k\eta}}) = \left(\frac{1}{B}\sum_{b=1}^{B}\nabla_a Q^{W_{k\eta}}(s_b, a_b, t_b)\Phi(s_b; W_0) - \nabla_W J(W)|_{W=W_{k\eta}}\right)$. Therefore, we have $\mathbb{E}_{\mathbb{B}_W}[\xi(W, \mathbb{B}_W)] = 0$ given an unbiased sampling mechanism for $\mathbb{B}_W$. Simiar formulation of SGD is also used in supervised learning (Cheng et al., 2020; Ben Arous et al., 2022).

## 4 Main Result: Locally Attainable States

The state space is typically thought of as a dense Euclidean space with all states reachable, but it is not necessarily the case that all such states are reachable by the agent. Three main factors constrain the states available to an agent: **1)** the transition function, **2)** the family of functions to which the policy belongs, and **3)** the optimization process which determines the dynamics of parameters of the policies. We therefore are interested in the set of states *attained* by the trajectories of linearised policy with parameters that are optimised as in Section 3.2 around a fixed state $s$ for time $\delta$. The properties of this set gives us a proxy for the "local manifold" around any arbitrary state.

A vector field, its exponential map, and the corresponding Lie series described in Section 2.3 are analogous to parameterised policy, the state transition based on this policy, and an approximation of this rollout. To formalise this, we denote the vector field determined by the parameters $W$ of a linearised policy with initialisation $W^0$ is

$$X(W) = g(x) + h(x, \Phi(x; W^0)W^0)\Phi(x; W^0)W. \tag{10}$$

The set of states attained by the rollout of this policy, parameterised by $W, W^0$, over time $\delta$ is therefore $e_{(0,\delta)}^{X(W)}(s)$, i.e. the image of the interval $(0, \delta)$ under the exponential map corresponding

to the vector field $X(W, W_0)$. Moreover, $W_0$ is randomly initialised, and the parameters $W$ are obtained through stochastic semigradient updates (equation 9).

There are two time scales: one is the time of policy rollouts and the other is the policy parameter optimisation. This complicates the analysis. We will use $t$ for time in the *physical* sense of an RL environment and $\tau$ for the gradient update step. Continuous-time analogues for discrete stochastic gradient descent algorithms at small step sizes have yielded remarkable theoretical analyses of algorithms (Mei et al., 2018b; Chizat & Bach, 2018; Jacot et al., 2018; Lee et al., 2019; Cheng et al., 2020; Ben Arous et al., 2022). Therefore, to analyse the evolution of the attainable states under a time-discretized sequence of parameters, we derive an approximate continuous time dynamics for the evolution of the randomly initialised parameters $W$. Many theoretical frameworks that study SGD in continuous time seek to approximate the evolution of the high-dimensional parameter distribution, but we seek to closely approximate the Lie series. We therefore utilise the theoretical framework provided by Ben Arous et al. (2022), with appropriate modifications, to analyse continuous-time dynamics of relevant statistics in the infinite-width limit.

Let $\xi_n$, $G_n$ be the semi-gradients for linearised policy of width $n$, $f_n^{\text{lin}}$. Let $\eta_n$ be a sequence of learning rates such that $\eta_n \to 0$ as $n \to \infty$ at rate $\frac{1}{\sqrt{n}}$. For a random variable $\mathbf{W}_n$, which determines the distribution of the $n d_s$ parameters, let $e_t^{X(\mathbf{W}_n)}(s)$ denote the push-forward of $\mathbf{W}_n$ of the exponential map. In the case of random variables, the attained set of states is sampled from this time-dependent push-forward of the distribution $\mathbf{W}_n^\tau$, where $\tau$ is the gradient time step. We make the following assumptions.

**Assumption 4.** *Suppose $H_n(W, \mathbb{B}_W) = \xi_n(W, \mathbb{B}_W) - G_n(W)$ for any $n$ and a given compact set $K$ there exists a constant $\sigma_{H,K}$ such that $\mathbb{E}_{\mathbb{B}_W}\left[L^2(H_n(W, \mathbb{B}_W))^4\right] \leq n\sigma_{H,K}^2$ for $W \in K$, where $L^2$ is the 2-norm.*

This assumption is a relaxed version of the assumption on the variance of the gradient update (assumption 4.4) made by Wang et al. (2019). We make a further assumption about the Lipschitz continuity of $H_n$ and $G_n$, similar to Ben Arous et al. (2022).

**Assumption 5.** *$G_n$ is locally Lipschitz continuous in $W$.*

Furthermore, we assume that the activation, $\varphi$, has bounded first and second derivatives everywhere in $\mathbb{R}$. This assumption holds for GeLU activation. We also denote by $Jh_j(s)$ the $d_s \times d_s$ Jacobian of the $d_s \times 1$ vector-valued function $h_j(s)$. We also define the proximity of a random variable to a manifold in a probabilistic manner.

**Definition 6.** *A random variable, $X$, is concentrated around a manifold $\mathcal{M}$ with a rate $\mathcal{R}$ if $\Pr(distance(X, \mathcal{M}) \geq D - O(\epsilon)) \leq e^{-\mathcal{R}(D)}$.*

Intuitively, this means that the probability that the random variable $X$ lies at more distance than $D$ decays exponentially.

> **Theorem 1.** *Given a continuous time MDP $\mathcal{M}$, a fixed state $s$, a sequence of two-layer linearised neural network policy, $f_n^{lin}$, initialised with i.i.d samples from $Normal(0, 1/d_s)$, semi-gradient based updates $(\eta_n, \xi_n, G_n)$ which satisfy assumptions 4, 5, then for varying $\delta t \in (0, \delta)$ and fixed $\tau > 0$ the random variable defined by the push-forward of the random variable $W_n^\tau$ w.r.t the exponential map $e_{\delta t}^{X(W_n^\tau)}(s)$ converges weakly to a random variable $\hat{S} + \bar{S}$ such that $\hat{S}$ concentrates around an $m$-dimensional manifold $M_{\delta', \tau}$ with $m \leq 2d_a + 1$ at rate $\mathcal{R}$, that depends on the operator norms of the matrices $Jh_j(s), j \in \{1, \ldots, d_a\}$, the values $g_k(s), k \in \{1, \ldots, d_s\}, \tau$. Further, the variable $\bar{S}$ is $O(\delta^3)$.*

Intuitively, this means that in the infinite width limit for very low learning rates the probability mass of the push-forward of the exponential map is concentrated around a $2d_a + 1$ dimensional manifold and this probability decays exponentially as one moves away from this manifold. The proof is provided in Appendix G. The proof sketch is as follows:

1. We expand the Lie series up to an error term of $\delta^3$ (Appendix B).

2. We then show the weak convergence of the dynamics of random variables that determine the Lie series in Appendix F, this Section closely follows the proof by Ben Arous et al. (2022).

3. Finally, we show that the push forward of the random variable $W^\tau$ through Lie series expansion is concentrated around a space spanned by $2d_a + 1$ vectors for fixed $t$ and therefore for variable $t$ there is a $2d_a + 2$ around which the data lie, modulo the $\delta^3$ distance.

The manifold $\mathcal{M}$ is locally derived as being spanned by $(h_j, v_j^\tau, tg + t^2 g')$ locally at $s$. Here, the directions in which the individual action dimensions change the state locally are $h_1, \ldots, h_{d_a}$. The mean second-order change: $v_j^\tau = \sum_{k=1}^{d_s} \frac{\partial h_j(s)}{\partial s_k} \sum_{j'=1}^{d_a} \bar{a}_{j'}^\tau h_{j',k}(s)$, where $Jh_j$ is the Jacobian of the function $h_j(s)$ and $\bar{a}_j^\tau$ is a constant that depends on the gradient time $\tau$, and the paraboloid. $g'$ is first order partial derivative of $g$, and therefore $tg + t^2 g'$ is a paraboloid. This is similar to how a local neighborhood is defined as being spanned by bases vectors in Section 2.2. Informally extending and intuiting this result, one can hypothesize that over the training dynamics of a linearised neural network, if the parameters remain bounded, the union of trajectories starting from a state $s$ over a "small" time interval $\delta$ then the trajectories are concentrated around a $2d_a + 3$ manifold. The reason being that their is an additional degrees of freedom from the gradient dynamics. This means "locally" the data is concentrated around some low-dimensional manifold whose dimensionality is linear in $d_a$.

## 5 EMPIRICAL VALIDATION

Our empirical validation is threefold. First, we show the validity of the linearised parameterisation of the policy (Equation 7) as a theoretical model for canonical NNs (Equation 6). Second, we verify that the bound on the manifold dimensionality as in Theorem 1 holds in practice. In the third subsection, we demonstrate the practical relevance of our result by demonstrating the benefits of learning compact low-dimensional representations, without significant computational overhead.

### 5.1 APPROXIMATION ERROR WITH LINEARISED POLICY

We empirically observe the impact of our choice of linearised policies as a theoretical model for two-layer NNs by measuring the impact on the returns of this choice. We calculate the difference in returns for DDPG using canonical NNs and linearised NNs as parameterisations for its policy network, while only training the weights of the first layer. Let the empirically observed return to which the DDPG algorithm converges using a canonical NN policy be $J_n^*$, and $J_n^{\mathrm{lin}}$ be the same for a linearised policy. In figure 2 we report the value $(J_n^* - J_n^{\mathrm{lin}})$ on the y-axis and $\log_2 n$ on the x-axis for the Cheetah environment (Todorov et al., 2012; Brockman et al., 2016). We present additional training curves in the appendix (figure 7) that compare how the returns vary as training progresses. In-

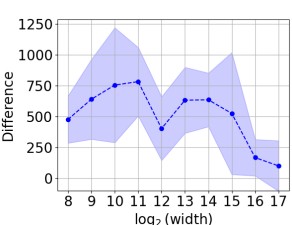

Figure 2: We observe that the difference between the returns approaches zero as we increase the width.

terestingly, at large widths ($\log_2 n > 15$) the discounted returns match across training steps for canonical and linearised policy parameterisations. This suggests that the agent's learning dynamics are captured by a linearised policy as $n \to \infty$. All results are averaged across 16 seeds.

### 5.2 EMPIRICAL DIMENSIONALITY ESTIMATION

We empirically corroborate our main result (Theorem 1) in the MuJoCo domains provided in the OpenAI Gym (Brockman et al., 2016). These are all continuous state and action spaces with $d_a < d_s$ for simulated robotic control tasks. The states are typically sensor measurements such as angles, velocities, or orientation, and the actions are torques provided at various joints. We estimate the dimensionality of the attainable set of states upon training. To sample data from the manifold, we record the trajectories of multiple DDPG evaluation runs across different seeds (Lillicrap et al., 2016), with two changes: we use GeLU activation (Hendrycks & Gimpel, 2016) instead of ReLU in both policy and value networks, and we also use a single hidden layer network instead of 2 hidden layers for both networks. Performance is comparable to the original DDPG architecture (see Appendix L). For background on DDPG refer to Appendix J. These choices keep our evaluation of

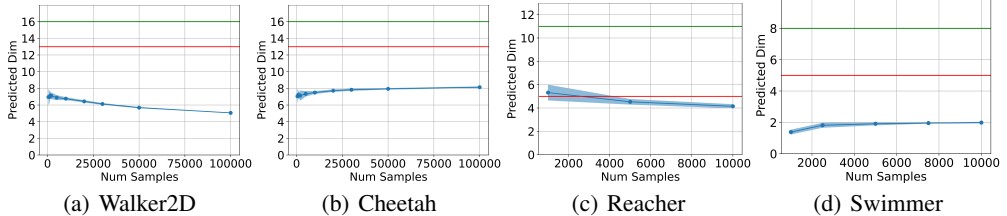

| (a) Walker2D | (b) Cheetah | (c) Reacher | (d) Swimmer |

Figure 3: Estimated dimensionality of the attainable states, in blue, is far below $d_s$ (green line) and also below $2d_a + 1$ (red line) for four tasks, estimated using the method by Facco et al. (2017).

the upper bound as close to the theoretical assumptions as possible while still resulting in reasonably good discounted returns. We then randomly sample states from the evaluation trajectories to obtain a subsample of states, $\mathcal{D} = \{s_i\}_{i=1}^n$. We estimate the dimensionality with 10 different subsamples of the same size to provide confidence intervals.

We employ the dimensionality estimation algorithm introduced by Facco et al. (2017), which estimates the intrinsic dimension of datasets characterized by non-uniform density and curvature, to empirically corroborate Theorem 1. More details on the dimensionality estimation procedure are presented in the Appendix I. Estimates for four MuJoCo environments are shown in Figure 3. For all environments, the estimate remains below the limit of $2d_a + 1$ in accordance with Theorem 1.

## 5.3 EMPIRICAL VALIDATION IN TOY LINEAR ENVIRONMENT

A deterministic system is fully reachable if given any start state, $s_0 \in \mathbb{R}^{d_s}$, the system can be driven to any goal state in $\mathbb{R}^{d_s}$. To contrast our result to classic control theory, we demonstrate that for a control environment which is fully reachable using a time-variant or open loop policy the set of all the attainable states using a bounded family of linearised neural nets (definition 3) is low-dimensional. A common example of a fully reachable $d_s$-dimensional linear control problem with 1D controller is:

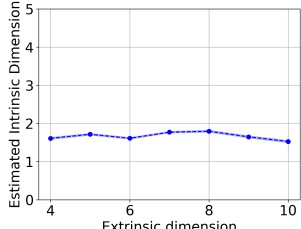

$$\dot{s}(t) = \begin{bmatrix} 0 & 1 & 0 & \dots & 0 \\ 0 & 0 & 1 & \dots & 0 \\ \vdots & \vdots & \vdots & \vdots & \vdots \\ 0 & 0 & 0 & \dots & 0 \end{bmatrix} s(t) + \begin{bmatrix} 0 \\ 0 \\ \vdots \\ 1 \end{bmatrix} \pi(t), \quad (11)$$

Figure 4: The intrinsic dimensionality estimate of attainable states linear fully reachable system under linearised policy on y-axis.

which is fully reachable. This follows from the fact that a linear system $\dot{x} = Ax + Bu(t)$ is fully reachable if and only if its controllability matrix defined by $\mathcal{C} = [B, AB, A^2B, \dots, A^{d_s-1}B]$ is full rank (Kalman, 1960; Jurdjevic, 1997). We instead evaluate the intrinsic dimension of the locally attainable set under feedback policies within our theoretical framework. We do so for the set of states attained for small $t$ under the dynamics $\dot{s}(t) = As(t) + B\Phi(x)W$, where $A, B$ are as in equation 11. To achieve this, for a fixed embedding dimension $d_s$ we obtain neural networks sampled uniformly randomly from the family of linearised neural networks as in definition 3, with $r = 1.0, t \in (0, 5), n = 1024$. Consequently, we obtain 1000 policies with $\delta t = 0.01$, and therefore a sample of 500000 states to estimate the intrinsic dimension of the attained set of states using the algorithm of Facco et al. (2017). We vary the dimensionality of the state space, $d_s$, from 3 to 10 to observe how the intrinsic dimension of the attained set of states varies with the embedding dimension while keeping $d_a$ fixed at 1. The dimensionality of the attained set of states remains upper-bounded by $2d_a + 1 = 3$ for this system (figure 4). This bound is even lower (at $d_a + 1 = 2$) for linear environments because the Lie series expansion (equation 4) gets truncated at $l = 1$ for GeLU activation owing to the fact that the second derivative is close to zero in most of $\mathbb{R}$.

## 5.4 REINFORCEMENT LEARNING WITH LOCAL LOW-DIMENSIONAL SUBSPACES

To demonstrate the applicability of our theoretical result, we apply a fully-connected *sparisification* MLP layer introduced by Yu et al. (2023a). In a series of works named the CRATE frame-

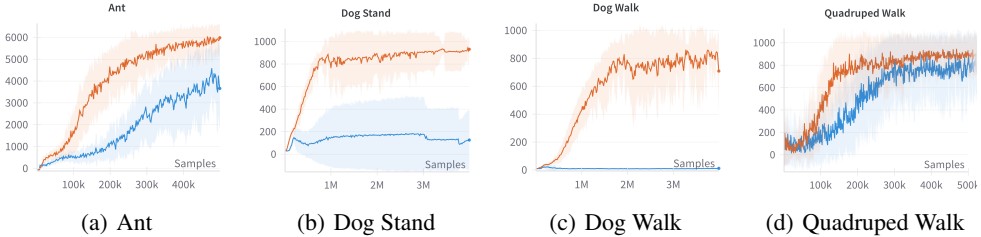

(a) Ant      (b) Dog Stand      (c) Dog Walk      (d) Quadruped Walk

Figure 5: Discounted returns of SAC (blue) and sparse SAC (red) $\alpha_\pi = 0.5, \alpha_Q = 0.4$

work (Coding RAte reduction TransformEr) (Chan et al., 2022; Yu et al., 2023a;b; Pai et al., 2024), researchers have argued for a better design of neural networks that compress and transform high-dimensional data given that it is sampled from low-dimensional manifolds. They assumed that the data lie near a union of low-dimensional manifolds $\cup_i M_i$, where each manifold has dimension $d_i \ll d_s$. An innovation that has remarkable empirical and theoretical results under the manifold hypothesis learns *sparse* high-dimensional representations of the data $\phi : \mathbb{R}^{d_s} \to \mathbb{R}^n$ with $n \geq \sum_i d_i$. These representations are orthogonal for data points across two manifolds, $x_i \in M_i, x_j \in M_j, i \neq j \implies \phi(s_i) \cdot \phi(s_j) = 0$, and low-rank on or near the same manifold, $\operatorname{rank}([\phi(x_i^1), \phi(x_i^2), \ldots, \phi(x_i^k)]) \approx d_i$ for $x_i^j \in M_i$. This can be viewed as disentangling representations across different manifolds via sparsification. The sparsification layer of width $n$ (Yu et al., 2023a) is defined as

$$Z^{\ell+1} = \operatorname{ReLU}\left(Z^\ell + \alpha W^\intercal \left(Z^\ell - W Z^\ell\right) - \alpha\lambda_1\right), \tag{12}$$

where $\alpha$ is the *sparse rate step size* parameter, $Z^\ell$ is the input to the $\ell$-th layer, $W$ are the $n \times n$ weight matrix. For further explanation of this sparsification layer, refer to Appendix M. As is evident, this is a linear transformation of the feed-forward layer and therefore does not add computational overhead. To verify the efficacy of disentangled low-dimensional representations, under the manifold hypothesis, we replace one feed-forward layer of all the policy and Q networks with a sparsification layer within the SAC framework (see Appendix K for background on SAC). We also use wider networks of width 1024, for both the baseline and modified architecture, for comparison. This is to satisfy the assumption $n \geq \sum_i d_i$ described above. With a simple code change of about 5 lines with same number of parameters and two additional hyperparameters, $\alpha_\pi, \alpha_Q$, we see improvements in the discounted returns for high-dimensional control environments: Ant (Brockman et al., 2016), Dog Stand, Dog Walk and Quadruped Walk (Tunyasuvunakool et al., 2020), averaged across 16 seeds. Discounted returns are reported on the y-axis against the number of samples on the x-axis in Figure 5. We observe that SAC with fully connected network fails to learn in high-dimensional Dog environments where as SAC equipped with a single sparsification layer instead of a fully connected layer does far better. This demonstrates the efficacy of learning local low-dimensional representations which arise from wide neural nets. We use the same hyperparameter for learning rates and entropy regularization for both the sparse SAC and vanilla SAC as those provided in the CleanRL library (Huang et al., 2022). We report ablation for Ant and Humanoid domains over the step size parameter in Appendix N.

## 6  DISCUSSION

We have proved that locally there exists a low-dimensional structure to the continuous time trajectories of policies learned using a semi-gradient ascent method. We develop a theoretical model where both transition dynamics and training dynamics are continuous-time. Ours is not only the first result of its kind, but we also introduce new mathematical models for the study of RL. In addition, we exploit this low-dimensional structure for efficient RL in high-dimensional environments with minimal changes. For detailed related work, refer to Appendix O and address the broader applicability of our theoretical work in Appendix P. We also assume access to the true value function $Q$, this is not practical and warrants an extension to the setting where this function is noisy. A key challenge that remains is extending this theory to very high-dimensional datasets where $d_s \to \infty$ as $n \to \infty$. We anticipate that noise in this settings will further complicate analysis. Additionally, the impact of stochastic transitions remains unexplored, as our current analysis assumes deterministic transitions.

## 7 ACKNOWLEDGMENTS

This research was partially funded by the ONR under the REPRISM MURI N000142412603 and ONR #N00014-22-1-2592. Partial funding for this work was provided by The Robotics and AI Institute, for which we are very grateful. This research was conducted using computational resources and services at the Center for Computation and Visualization, Brown University. We thank Sam Lobel, Rafael Rodriguez Sanchez, Akhil Bagaria, and Tejas Kotwal for fruitful conversations and guidance on the project. This work would not have been possible without their input. We would like to thank the members of Brown Intelligent Robot Lab for their continued support. We thank Wasiwasi Mgonzo, Alessio Mazzetto, Ji Won Chung, Renato Amado, Pedro Lopes De Almeida, Kalaiyarasan Arumugam, the second floor community at the Center for Information Technology, and the larger Brown University graduate student community for their moral support and encouragement. The anonymous reviewers who have contributed very significantly to our work over time have our sincere gratitude.

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

# Appendix

## A  MANIFOLD BACKGROUND

Here, we provide precise definitions which are essential to the theory of differential geometry but might not be absolutely essential to understanding our results in the main body of our work. The tangent space characterises the geometry of the manifold and it is defined as follows.

**Definition 7.** *Let $M$ be an $m$-manifold in $\mathbb{R}^k$ and $p \in M$ be a fixed point. A vector $v \in \mathbb{R}^k$ is called a tangent vector of $M$ at $p$ if there exists a smooth curve $\gamma : I \to M$ such that $\gamma(0) = p, \dot{\gamma}(0) = v$. The set $T_p M := \{\dot{\gamma}(0) | \gamma : \mathbb{R} \to M$ is smooth, $\gamma(0) = p\}$ of tangent vectors of $M$ at $p$ is called the tangent space of $M$ at $p$.*

Continuing our example, the tangent space of a point $p$ in $S^2$ is the vertical plane tangent to the cylinder at that point. For a small enough $\epsilon$ and a vector $v \in T_p S^2$ there exists a unique curve $\gamma : [-\epsilon, \epsilon] \to S^2$ such that $\gamma(0) = p$ and $\dot{\gamma}(0) = v$. The union of tangent spaces at all points is termed the tangent bundle and denoted by $T(M)$. At point $p$, the tangent space $T_p M$ is spanned by the vectors $\left\{ \frac{\partial}{\partial x^i} \Big|_p \right\}$. Any tangent vector $v \in T_p M$ can be expressed as a linear combination:

$$v = \sum_{i=1}^{m} v^i \frac{\partial}{\partial x^i} \Big|_p,$$

where $v^i \in \mathbb{R}$ are the components of $v$ in the basis $\left\{ \frac{\partial}{\partial x^i} \Big|_p \right\}$.

## B  FEEDBACK ACTION LIE SERIES

Consider the vector fields for a feedback policy $a(x) \in C^\infty$:

$$X = g(x) + h(x)a(x).$$

Consider the Lie series and its first term:

$$e_t^X = x + tX(x) + \sum_{l=1}^{\infty} \frac{t^{l+1}}{(l+1)!} L_X^{l+1}(x)$$

$$L_{X_1}(x) = g(x) + h(x)a(x).$$

The second order term can be written as:

$$
\begin{aligned}
(L_{X_1}^2 x)_i &= \sum_{k=1}^{d_s} \left( g_k(x) + \sum_{j=1}^{d_a} h_{j,k}(x)a_j(x) \right) \frac{\partial g_i(x) + h_i(x)a(x)}{\partial x_k} \\
&= \sum_{k=1}^{d_s} \left( g_k(x) \frac{\partial g_i(x)}{\partial x_k} + \sum_{j=1}^{d_a} h_{j,k}(x)a_j(x) \frac{\partial g_i(x)}{\partial x_k} \right. \\
&\quad + g_k(x) \left( \sum_{j'=1}^{d_a} \frac{\partial h_{i,j'}(x)}{\partial x_k} a_{j'}(x) + \frac{\partial a_{j'}(x)}{\partial x_k} h_{i,j'}(x) \right) \\
&\quad \left. + \sum_{j=1}^{d_a} \sum_{j'=1}^{d_a} h_{j,k}(x)a_j(x) \frac{\partial h_{i,j'}(x)}{\partial x_k} a_{j'}(x) + h_{j,k}(x)a_j(x) \frac{\partial a_{j'}(x)}{\partial x_k} h_{i,j'}(x) \right).
\end{aligned}
$$

While we do not use this third order term we present it to demonstrate that tracking the statistics of this gets exponentially difficult:

$$
\begin{aligned}
(L_{X_1}^3 x)_i &= \sum_{k'=1}^{d_s} \left( g_k(x) + \sum_{j=1}^{d_a} h_{j,k}(x) a_j(x) \right) \frac{(L_{X_1}^2 x)_i}{\partial x_{k'}} \\
&= \sum_{k'=1}^{d_s} \left( g_k'(x) + \sum_{j''=1}^{d_a} h_{j'',k'}(x) a_{j''}(x) \right) \left( \sum_{k=1}^{d_s} \frac{\partial g_k(x)}{\partial x_{k'}} + g_k(x) \frac{\partial g_i(x)}{\partial x_{k'} \partial x_k} \right. \\
&\quad + \sum_{j=1}^{d_a} \frac{\partial h_{j,k}(x)}{\partial x_{k'}} a_j(x) \frac{\partial g_i(x)}{\partial x_k} + h_{j,k}(x) \frac{\partial a_j(x)}{\partial x_{k'}} \frac{\partial g_i(x)}{\partial x_k} + h_{j,k}(x) a_j(x) \frac{\partial^2 g_i(x)}{\partial x_{k'} \partial x_k} \\
&\quad + \sum_{j'=1}^{d_a} \frac{\partial g_k(x)}{\partial x_{k'}} \frac{\partial h_{i,j'}(x)}{\partial x_k} a_{j'}(x) + g_k(x) \frac{\partial^2 h_{i,j'}(x)}{\partial x_{k'} \partial x_k} a_{j'}(x) + g_k(x) \frac{\partial h_{i,j'}(x)}{\partial x_k} \frac{\partial a_{j'}(x)}{\partial x_{k'}} \\
&\quad + \sum_{j'=1}^{d_a} \frac{\partial g_k(x)}{\partial x_{k'}} \frac{\partial a_{j'}(x)}{\partial x_k} h_{i,j'}(x) + g_k(x) \frac{\partial^2 a_{j'}(x)}{\partial x_{k'} \partial x_k} h_{i,j'}(x) + g_k(x) \frac{\partial a_{j'}(x)}{\partial x_k} \frac{\partial h_{i,j'}(x)}{\partial x_{k'}} \\
&\quad + \sum_{j=1}^{d_a} \sum_{j'=1}^{d_a} \left( \frac{\partial h_{j,k}(x)}{\partial x_{k'}} a_j(x) \frac{\partial h_{i,j'}(x)}{\partial x_k} a_{j'}(x) + h_{j,k}(x) \frac{\partial a_j(x)}{\partial x_{k'}} \frac{\partial h_{i,j'}(x)}{\partial x_k} a_{j'}(x) + h_{j,k}(x) a_j(x) \frac{\partial^2 h_{i,j'}(x)}{\partial x_{k'} \partial x_k} a_{j'}(x) \right. \\
&\quad + h_{j,k}(x) a_j(x) \frac{\partial h_{i,j'}(x)}{\partial x_k} \frac{\partial a_{j'}(x)}{\partial x_{k'}} + \frac{\partial h_{j,k}(x)}{\partial x_{k'}} a_j(x) \frac{\partial a_{j'}(x)}{\partial x_k} h_{i,j'}(x) + h_{j,k}(x) \frac{\partial a_j(x)}{\partial x_{k'}} \frac{\partial a_{j'}(x)}{\partial x_k} h_{i,j'}(x) \\
&\quad \left. \left. + h_{j,k}(x) a_j(x) \frac{\partial^2 a_{j'}(x)}{\partial x_{k'} \partial x_k} h_{i,j'}(x) + h_{j,k}(x) a_j(x) \frac{\partial a_{j'}(x)}{\partial x_k} \frac{\partial h_{i,j'}(x)}{\partial x_{k'}} \right) \right).
\end{aligned}
$$

## C  SOME HELPFUL DERIVATIONS

Here we derive various expressions to bound their magnitude in terms of the width of the NN $n$. First we consider the gradient term:

$$
\begin{aligned}
G(Y) &= \lim_{N' \to \infty} \frac{1}{N'} \sum_{i=1}^{N'} \nabla_a \hat{Q}^Y(s_i, a_i, t_i) \Phi(s_i; W_0) \\
&= \lim_{N' \to \infty} \frac{\eta}{N'} \sum_{i=1}^{N'} q^i(s_i) \Phi(s_i; W_0) \\
&= \lim_{N' \to \infty} \frac{1}{N' \sqrt{n}} \sum_{i=1}^{N'} \left[ s_k^i \varphi'(W_m^0 \cdot s^i) \sum_{j=1}^{d_a} q_j^i(s_i) C_{j,m}^0 \right]_{m(d_s-1)+k} \\
&= \frac{1}{\sqrt{n}} \left[ \sum_{j=1}^{d_a} C_{j,m}^0 \lim_{N' \to \infty} \frac{1}{N'} \sum_{i=1}^{N'} s_k^i \varphi'(W_m^0 \cdot s^i) q_j^i(s_i) \right]_{m(d_s-1)+k} \\
&= \frac{1}{\sqrt{n}} \left[ \sum_{j=1}^{d_a} C_{j,m}^0 G'_{j,m(d_s-1)+k}(Y) \right]_{m(d_s-1)+k}.
\end{aligned}
$$

The fourth equality suggests that individual elements of the vector $G$ at any point are iid samples from the same random variable. Similarly we expand the term for $M^2(Y)$

$$
\begin{aligned}
\xi(Y) =& \frac{\sqrt{\eta}}{\sqrt{n}} \mathbb{E}_{\mathbb{B}_Y} \left[ \frac{1}{B} \sum_{b=1}^{B} \sum_{j=1}^{d_a} q_j(s^b) C_{j,m}^0 \varphi'(W_m^0 \cdot s^b) s_k^b - \sum_{j=1}^{d_a} C_{j,m}^0 G'_{j,m(d_s-1)+k}(Y) \right]_{m(d_s-1)+k} \\
=& \frac{\sqrt{\eta}}{\sqrt{n}} \mathbb{E}_{\mathbb{B}_Y} \left[ \sum_{j=1}^{d_a} C_{j,m}^0 \left( \frac{1}{B} q_j(s^b) \sum_{b=1}^{B} \varphi'(W_m^0 \cdot s^b) s_k^b - \sum_{j=1}^{d_a} G'_{j,m(d_s-1)+k}(Y) \right) \right]_{m(d_s-1)+k} \\
=& \frac{\sqrt{\eta}}{\sqrt{n}} \sum_{j=1}^{d_a} \left[ C_{j,m}^0 \mathbb{E}_{\mathbb{B}_Y} \left[ \frac{1}{B} q_j(s^b) \sum_{b=1}^{B} \varphi'(W_m^0 \cdot s^b) s_k^b - \sum_{j=1}^{d_a} G'_{j,m(d_s-1)+k}(Y) \right] \right]_{m(d_s-1)+k} \\
M^2(Y) =& \frac{\eta}{n} \mathbb{E}_{\mathbb{B}_Y} \left[ \left( \frac{1}{B} \sum_{b=1}^{B} \sum_{j=1}^{d_a} q_j(s^b) C_{j,m}^0 \varphi'(W_m^0 \cdot s^b) s_k^b - \sum_{j=1}^{d_a} C_{j,m}^0 G'_{j,m(d_s-1)+k}(Y) \right) \right. \\
& \left. \left( \frac{1}{B} \sum_{b'=1}^{B} \sum_{j'=1}^{d_a} q_{j'}(s^{b'}) C_{j',m'}^0 \varphi'(W_{m'}^0 \cdot s^{b'}) s_{k'}^{b'} - \sum_{j'=1}^{d_a} C_{j',m'}^0 G'_{j',m'(d_s-1)+k'}(Y) \right) \right]_{m(d_s-1)+k,m'(d_s-1)+k'} \\
=& \frac{\eta}{n} \mathbb{E}_{\mathbb{B}_Y} \left[ \left( \frac{1}{B^2} \sum_{b'=1}^{B} \sum_{j'=1}^{d_a} \sum_{b=1}^{B} \sum_{j=1}^{d_a} q_j(s^b) q_{j'}(s^{b'}) C_{j,m}^0 \varphi'(W_m^0 \cdot s^b) s_k^b C_{j',m'}^0 \varphi'(W_{m'}^0 \cdot s^{b'}) s_{k'}^{b'} \right) \right. \\
& - \left( \frac{1}{B} \sum_{j'=1}^{d_a} \sum_{b=1}^{B} \sum_{j=1}^{d_a} q_j(s^b) C_{j,m}^0 \varphi'(W_m^0 \cdot s^b) s_k^b C_{j',m'}^0 G'_{j',m'(d_s-1)+k'}(Y) \right) \\
& - \frac{1}{B} \sum_{j=1}^{d_a} \sum_{b'=1}^{B} \sum_{j'=1}^{d_a} C_{j,m}^0 G'_{j,m(d_s-1)+k}(Y) q_{j'}(s^{b'}) C_{j',m'}^0 \varphi'(W_{m'}^0 \cdot s^{b'}) s_{k'}^{b'} \\
& \left. + \sum_{j=1}^{d_a} \sum_{j'=1}^{d_a} C_{j,m}^0 G'_{j,m(d_s-1)+k}(Y) C_{j',m'}^0 G'_{j',m'(d_s-1)+k'}(Y) \right]_{m(d_s-1)+k,m'(d_s-1)+k'},
\end{aligned}
$$

which we combine to form:

$$
\begin{aligned}
M^2(Y) =& \frac{\eta}{n} \left[ \sum_{j=1}^{d_a} \sum_{j'=1}^{d_a} C_{j,m}^0 C_{j',m'}^0 \mathbb{E}_{\mathbb{B}_Y} \left[ \frac{1}{B^2} \sum_{b'=1}^{B} \sum_{b'=1}^{B} q_j(s^b) \varphi'(W_m^0 \cdot s^b) s_k^b q_{j'}(s^{b'}) \varphi'(W_{m'}^0 \cdot s^{b'}) s_{k'}^{b'} \right. \right. \\
& - \left( \frac{1}{B} \sum_{b=1}^{B} q_{j'}(s^b) C \varphi'(W_m^0 \cdot s^b) s_k^b G'_{j',m'(d_s-1)+k'}(Y) \right) \\
& - \left( \frac{1}{B} \sum_{b=1}^{B} q_j(s^b) \varphi'(W_{m'}^0 \cdot s^b) s_k^b G'_{j,m(d_s-1)+k}(Y) \right) \\
& \left. \left. + G'_{j,m(d_s-1)+k}(Y) G'_{j',m'(d_s-1)+k'}(Y) \right] \right]_{m(d_s-1)+k,m'(d_s-1)+k'}.
\end{aligned}
$$

Let the internal term in the summation be defined as follows:

$$
\begin{aligned}
H^{j,j'}_{m(d_s-1)+k,m'(d_s-1)+k'} =& \mathbb{E}_{\mathbb{B}_Y}\Bigg[ \frac{1}{B^2} \sum_{b=1}^{B}\sum_{b'=1}^{B} q_j(s^b)\varphi'(W_m^0\cdot s^b)s_k^b q_{j'}(s^{b'})\varphi'(W_{m'}^0\cdot s^{b'})s_{k'}^{b'} \\
&- \left( \frac{1}{B}\sum_{b=1}^{B} q_j(s^b)C\varphi'(W_m^0\cdot s^b)s_k^b G'_{j',m'(d_s-1)+k'}(Y) \right) \\
&- \left( \frac{1}{B}\sum_{b=1}^{B} q_{j'}(s^b)\varphi'(W_{m'}^0\cdot s^b)s_k^b G'_{j,m(d_s-1)+k}(Y) \right) \\
&+ G'_{j,m(d_s-1)+k}(Y)G'_{j',m'(d_s-1)+k'}(Y)\Bigg] \\
=& \mathbb{E}_{\mathbb{B}_Y}\Bigg[ \left( \frac{1}{B}\sum_{b'=1}^{B} q_j(s^{b'})\varphi'(W_m^0\cdot s^{b'})s_k^{b'} - G'_{j,m(d_s-1)+k}(Y) \right) \\
&\left( \frac{1}{B}\sum_{b'=1}^{B} q_{j'}(s^{b'})\varphi'(W_{m'}^0\cdot s^{b'})s_k^{b'} - G'_{j',m'(d_s-1)+k'}(Y) \right)\Bigg]
\end{aligned}
$$

## D COVARIATE TERMS

We denote $\nabla_a Q^W(s_b, a, t_b)|_{a=a_b}$ by $[q_1(s^b),\ldots,q_{d_a}(s^b)]$ as shorthand. Consider the term:

$$
\begin{aligned}
\mathbb{E}\left[ M_l^{A_j} M_l^{A_{j'}} \right] =& \nabla_Y A_j(s) \cdot M^2(Y)\nabla_Y A_{j'}(s) \\
=& \Phi_j(s, W_0) \cdot M^2(Y)\Phi_{j'}(s, W_0) \\
=& \frac{1}{n^{3/2}}\Phi_j(s, W_0) \cdot \Bigg[ \sum_{m'=1}^{n}\sum_{k'=1}^{d_s}\sum_{l=1}^{d_a}\sum_{l'=1}^{d_a} C_{l,m}^0 C_{l',m'}^0 H^{l,l'}_{m'(d_s-1)+k,m(d_s-1)+k'}(Y) \\
& C_{j',m'}^0\varphi'(W_{m'}^0\cdot s)s_{k'} \Bigg]_{m(d_s-1)+k}.
\end{aligned}
$$

We further expand the dot product with the $nd_s \times 1$ vector $\Phi_j(s, W_0)$

$$
\begin{aligned}
\mathbb{E}\left[M_l^{A_j} M_l^{A_{j'}}\right] =& \frac{1}{3n^2} \sum_{m=1}^{n} \sum_{k=1}^{d_s} \sum_{m'=1}^{n} \sum_{k'=1}^{d_s} \sum_{l=1}^{d_a} \sum_{l'=1}^{d_a} C_{l,m}^0 C_{l',m'}^0 C_{j',m'}^0 C_{j,m}^0 \\
& H_{m'(d_s-1)+k, m(d_s-1)+k'}^{l,l'}(Y) \varphi'(W_{m'}^0 \cdot s) s_{k'} \varphi'(W_m^0 \cdot s) s_{k'} \\
=& \frac{1}{3n^2} \sum_{m=1}^{n} \sum_{k=1}^{d_s} \sum_{m'=1}^{n} \sum_{k'=1}^{d_s} \sum_{l=1}^{d_a} \sum_{l'=1}^{d_a} C_{l,m}^0 C_{l',m'}^0 C_{j',m'}^0 C_{j,m}^0 \\
& \varphi'(W_{m'}^0 \cdot s) s_{k'} \varphi'(W_m^0 \cdot s) s_k \\
& \mathbb{E}_{\mathbb{B}_Y \mathbb{B}_Y'}\left[ \frac{1}{B^2} \sum_{b=1}^{B} \sum_{b'=1}^{B} q_l(s^b) \varphi'(W_m^0 \cdot s^b) s_k^b q_{l'}(s^{b'}) \varphi'(W_{m'}^0 \cdot s^{b'}) s_{k'}^{b'} \right. \\
& - \left( \frac{1}{B} \sum_{b=1}^{B} q_l(s^b) C \varphi'(W_m^0 \cdot s^b) s_k^b G'_{l',m'(d_s-1)+k'}(Y) \right) \\
& - \left( \frac{1}{B} \sum_{b=1}^{B} q_{l'}(s^b) \varphi'(W_{m'}^0 \cdot s^b) s_k^b G'_{l,m(d_s-1)+k}(Y) \right) \\
& \left. + G'_{l,m(d_s-1)+k}(Y) G'_{l',m'(d_s-1)+k'}(Y) \right] \\
=& \frac{2}{3n^2} \mathbb{E}_{\mathbb{B}_Y \mathbb{B}_Y'}\left[ \sum_{m=1}^{n} \sum_{k=1}^{d_s} \sum_{m'=1}^{n} \sum_{k'=1}^{d_s} \varphi'(W_{m'}^0 \cdot s) s_{k'} \varphi'(W_m^0 \cdot s) s_k \right( \\
& (C_{j',m'}^0)^2 (C_{j,m}^0)^2 \frac{1}{B^2} \sum_{b=1}^{B} \sum_{b'=1}^{B} q_j(s^b) \varphi'(W_m^0 \cdot s^b) s_k^b q_{j'}(s^{b'}) \varphi'(W_{m'}^0 \cdot s^{b'}) s_{k'}^{b'} \\
& - (C_{j',m'}^0)^2 (C_{j,m}^0)^2 \frac{1}{B} \sum_{b=1}^{B} q_j(s^b) C \varphi'(W_m^0 \cdot s^b) s_k^b G'_{j',m'(d_s-1)+k'}(Y) \\
& - (C_{j',m'}^0)^2 (C_{j,m}^0)^2 \frac{1}{B} \sum_{b'=1}^{B} q_{j'}(s^{b'}) \varphi'(W_{m'}^0 \cdot s^{b'}) s_k^{b'} G'_{j,m(d_s-1)+k}(Y) \\
& \left. \left. + (C_{j',m'}^0)^2 (C_{j,m}^0)^2 G'_{j,m(d_s-1)+k}(Y) G'_{j',m'(d_s-1)+k'}(Y) \right) \right] \\
& + \sum_{l,l' \neq j,j'} \frac{2}{3n^2} M_{l,l'}^2.
\end{aligned}
$$

In the $n \to \infty$ we note that $\frac{2}{3n^2} M_{l,l',j,j'}^2 \to 0$ because we have $\mathbb{E}[C_l]\mathbb{E}[C_{l'}] \to 0$ as a multiplicative term, while the other terms are finite and bounded in second moment because of the boundedness

properties of gradient GeLU activation $\varphi'$. For $j = j'$ we have the following expression

$$
\begin{aligned}
\mathbb{E}\left[M_l^{A_j} M_l^{A_j}\right] = & \frac{2}{3n^2} \mathbb{E}_{\mathbb{B}_Y \mathbb{B}'_Y}\left[\sum_{m=1}^{n} \sum_{k=1}^{d_s} \sum_{m'=1}^{n} \sum_{k'=1}^{d_s} \varphi'(W_{m'}^0 \cdot s) s_{k'} \varphi'(W_m^0 \cdot s) s_k \left(\vphantom{\sum_{b=1}^{B}}\right.\right. \\
& (C_{j,m'}^0)^4 \frac{1}{B^2} \sum_{b=1}^{B} \sum_{b'=1}^{B} q_j(s^b) \varphi'(W_m^0 \cdot s^b) s_k^b q_j(s^{b'}) \varphi'(W_{m'}^0 \cdot s^{b'}) s_{k'}^{b'} \\
& - (C_{j,m'}^0)^4 \frac{1}{B} \sum_{b=1}^{B} q_j(s^b) C \varphi'(W_m^0 \cdot s^b) s_k^b G'_{j,m'(d_s-1)+k'}(Y) \\
& - (C_{j,m'}^0)^4 \frac{1}{B} \sum_{b'=1}^{B} q_j(s^{b'}) \varphi'(W_{m'}^0 \cdot s^{b'}) s_k^{b'} G'_{j,m(d_s-1)+k}(Y) \\
& \left.\left. + (C_{j,m'}^0)^4 G'_{j,m(d_s-1)+k}(Y)^2 \right)\right] + O\left(\frac{1}{n}\right),
\end{aligned}
\tag{13}
$$

where the $O\left(\frac{1}{n}\right)$ term is a result of the convergence rate of the strong law of large numbers (Vershynin, 2018).

## E   SUFFICIENT STATISTICS

We want to determine the dynamics of some linear or quadratic function of these parameters, for example the $j$-th output of the policy network $A_j(s; W) = \Phi_j(s; W_0)W$. Therefore, we find a setting where a continuous time SDE such as

$$
dA_j(s; W) = \mu(A_j(s; W))d\tau + \sigma(A_j(s; W))dw_\tau,
\tag{14}
$$

represent the dynamics of $A_j(s; W_{k\eta})$. In other words, we find the conditions under which $W_{k\eta}, Y_{k\eta}$ are close together in some sense.

Furthermore, we assume that the activation, $\varphi$, has bounded first and second derivatives almost everywhere in $\mathbb{R}$. This assumption holds for GeLU activation. Moreover, we would like to show that we can track the statistics corresponding to elements of the Lie series

$$
\begin{aligned}
A_j^\tau(s) &= f_j^{\text{lin}}(s; W^\tau), \\
A_{j,k}^\tau(s) &= \frac{\partial A_j^\tau(s)}{\partial x_k}, \\
A_{j,k,k'}^\tau(s) &= \frac{\partial^2 A_j^\tau(s)}{\partial x_{k'} \partial x_k}.
\end{aligned}
$$

The manner in which the push forward of the distribution of parameters described in Section 4, $e_t^{X(\mathbf{W}_n)}(s)$, changes with gradient steps is central to our work. In this Section we derive the sufficient statistics that determine how the distribution over actions and their quadratic combinations evolve over gradient steps. We present the following lemma for the learning setup described in Section 3.2 and the sufficient statistic required to track the changes in the action $A_j(s; W)$ as described

in Appendix E. First, we define the following random variables for a fixed $s \in \mathcal{S}$:

$$
\begin{aligned}
\hat{A}_j^n(s; W) &= \frac{1}{\sqrt{n}} \sum_{m=1}^{n} C_{j,m}^0 \varphi'(W_m^0 \cdot s)(W_m - W_m^0) \cdot s \\
&= \frac{1}{\sqrt{n}} \sum_{m=1}^{n} C_{j,m}^0 \varphi'(W_m^0 \cdot s)(W_m - W_m^0) \cdot s \\
&= \frac{1}{\sqrt{n}} \sum_{m=1}^{n} C_{j,m}^0 \varphi'(W_m^0 \cdot s) \Delta W_m \cdot s, \\
&= \frac{1}{\sqrt{n}} \sum_{m=1}^{n} C_{j,m}^0 \varphi'(W_m^0 \cdot s) \sum_{k=1}^{d_s} s_k \Delta W_{m(d_s-1)+k}, \\
&= \frac{1}{\sqrt{n}} \sum_{k=1}^{d_s} \sum_{m=1}^{n} C_{j,m}^0 \varphi'(W_{m(d_s-1)+k}^0 \cdot s) s_k \Delta W_{m(d_s-1)+k} \\
&= \sum_{k=1}^{d_s} Z_{j,k}(s; W)
\end{aligned}
$$

where $\Delta W = W - W^0$, $\Phi_j(s; W^0)$ is as defined in Section 3.1, and $C_{j,m}^0 \Delta W_{m,k} \sim Z_{j,k}(s; W)$ is the random variable that determines the action $j$ corresponding to the action $j$ and state space dimension $k$ given parameters $W$. This is because individual $m$ random parameters $\Delta W_{m,k} s_k$ can be viewed as i.i.d samples from a distribution (see Section C).

Similarly, we derive $\frac{\partial \hat{A}_j^\tau(s)}{\partial s_k}$ from the fifth equality above as follows:

$$
\frac{\partial \hat{A}_j^\tau(s)}{\partial x_k} = \frac{1}{\sqrt{n}} \sum_{k=1}^{d_s} \sum_{m=1}^{n} C_{j,m}^0 \Delta W_{m(d_s-1)+k} \frac{\partial \varphi'(W_{m(d_s-1)+k}^0 \cdot s) s_k}{\partial s_k},
$$

where once again the i.i.d copies of a random variable: $C_{j,m}^0 \Delta W_{m,k} \sim Z_{j,k}(s; W)$ appear in the expression . Therefore, to track the random variables corresponding to the quantities of interest in the first and second order Lie series in Section B we need to track $d_a d_s$ random variables: $Z_{j,k}$ with $j \in \{1, \ldots, d_a\}, k \in \{1, \ldots, d_s\}$.

## F  TRACKING STATISTICS

**Notation:** In this Section we use $x \lesssim y$ to denote that $x$ is less than $y$ times some constant. We also write $L_\infty^n(E_K^n)$ denoting the supremum of a function that depends on $n$ on the compact set $K$. As noted in Section E we aim track the following statistics for fixed $s$ across gradient steps

$$
A_j^n(s, W), A_j^n(s, W) A_{j'}^n(s, W), A_{j,k}^n(s, W).
$$

Moreover, we seek to derive their dynamics in the continuous-time limit. Given linearised parameterisation of a two-layer network policy (equation 7) and the gradient update is as described in Section 3.2. We present a lemma, whose proof follows the proof of Theorem 2.2 provided by Ben Arous et al. (2022), except that in our case the dimensionality of the input data remains constant, on the dynamics of summary statistics linear in the parameters that describe the learning dynamics under SGD. Here the sufficient statistics that determine the dynamics of the action $A_j^n$ is denoted by $\mathcal{X}_\tau$. We prove the dynamics of the $j$-th action.

**Lemma 8.** *Given a fixed state $s$ the $j$-th action, $A_j^n(s; \cdot)$, determined by a linearised neural policy with two hidden layers as described and initialised in Section 3, we assume $W_0 \sim \mathcal{X}_0 = Normal(0, I_{d_s}/d_s)$ i.i.d whose gradient dynamics are described in equation 8 with learning rates $\eta_n \to 0$, and under assumptions 4, 5, we have that in the limit $n \to \infty$ the dynamics of $A_j^n$ converge weakly to the following random ODE*

$$
d\bar{A}_j(s; \mathcal{X}_\tau) = v_j(s; \mathcal{X}_\tau) dt, \tag{15}
$$

*with the random variable $\mathcal{X}_\tau$ is the limit point of the sufficient staistics, $\mathcal{X}_t^n$, of the parameters updated according to stochastic policy gradient based updates laid out in Section 3.2 with $\eta = \eta_n$.*

*Proof.* Suppose the evolution of $W$ over gradient steps is as follows

$$W_\tau = W_{\tau-1} + \eta_n \xi_n(W, \mathbb{B}_{W_{\tau-1}}), \text{ where}$$

$$\xi_n(W, \mathbb{B}_{W_{\tau-1}}) = \frac{1}{B} \sum_{s_b, a_b \in \mathbb{B}_W} \nabla_a Q^W(s_b, a)|_{a=a_b} \Phi^n(s_b, a_b).$$

We further let $G_n(W_{\tau-1}) = \mathbb{E}_{\mathbb{B}_{W_{\tau-1}}} \left[ \xi_n(W, \mathbb{B}_{W_{\tau-1}}) \right]$. Let $H_n(W, \mathbb{B}_Y) = \xi(W, \mathbb{B}_{W_{\tau-1}}) - G_n(W)$. Further let

$$\Xi_n(W) = \mathbb{E}_{\mathbb{B}_W} \left[ H_n(W, \mathbb{B}_Y) H_n(W, \mathbb{B}_Y)^\intercal \right].$$

For the statistic $A_j^n$ consider the following evolution

$$\begin{aligned}
\nu_j^\tau - \nu_j^{\tau-1} &= \Phi_j^n G_n(W_\tau), \\
\varsigma_j^\tau - \varsigma_j^{\tau-1} &= \Phi_j^n H_n(W_\tau, \mathbb{B}_{W_\tau}),
\end{aligned} \tag{16}$$

where $\Phi_j^n$ represents the feature vector for the $j$-th action at state $s$. Omitting the subscript in $\eta_n$, since we take the limit $\eta_n \to 0$ as $n \to \infty$, we now consider the $u_j$ as follows,

$$u_j^\tau = u_j^0 + \eta \sum_{\tau'=1}^{\tau} (\nu_j^{\tau'} - \nu_j^{\tau'-1}) + \eta \sum_{\tau'=1}^{\tau'} (\varsigma_j^{\tau'} - \varsigma_j^{\tau'-1}).$$

Now for $l \in [0, L]$ we define,

$$\nu_j'(l) = \nu_j^{[l/\eta]} - \nu_j^{[l/\eta]-1}, \text{ and } \varsigma_j'(l) = \varsigma_j^{[l/\eta]} - \varsigma_j^{[l/\eta]-1}.$$

If we let

$$\mu_j^n(l) = \int_0^l \nu_j'(l) dl' = \nu_j'(\eta[l/\eta]) + (s - \eta[l/\eta]) \left( \nu_j^{[l/\eta]} - \nu_j^{[l/\eta]-1} \right)$$

$$\sigma_j^n(l) = \int_0^l \varsigma_j'(l) dl' = \varsigma_j'(\eta[l/\eta]) + (s - \eta[l/\eta]) \left( \varsigma_j^{[l/\eta]} - \varsigma_j^{[l/\eta]-1} \right),$$

be the continuous linear interpolations based on the discrete random variables $\nu, \varsigma$ and combine them together to obtain

$$v_j^n(l) = v_j^n(0) + \mu_j^n(l) + \sigma_j^n(l). \tag{17}$$

Given a compact set $K \subset \mathbb{R}$ and the exit time $\tau_K$ we aim to show that for all $0 \leq s, t \leq T$,

$$\mathbb{E}||v_j^n(s \wedge \tau_K) - v_j^n(t \wedge \tau_K)||^2 \lesssim_{K,T} (t-s)^4,$$

where the expectation over the stochastic updates. This proves that $v_j^n(s \wedge \tau_K)$ is 1/4 Hölder-continuous by Kolmogorov's continuity theorem (see Section 2.2 in the textbook by Karatzas & Shreve (2014)). We have for all $s, t$

$$||v_j^n(s) - v_j^n(t)|| \leq ||\mu_j^n(s) - \mu_j^n(t)|| + ||\sigma_j^n(s) - \sigma_j^n(t)||.$$

For a fixed $W$ the action $j$ corresponding to the linearised policy in the limit $n \to \infty$ is defined as:

$$\bar{a}_j(s, W) = \lim_{n \to \infty} \Phi_j^n(s) W^n.$$

Now further suppose $W$ is a stochastic variable where each of its entries is sampled i.i.d. from some distribution $\mathcal{X} \in \mathbb{P}(\mathbb{R})$, where $\mathbb{P}(\mathbb{R})$ is a probability space over $\mathbb{R}$ with the canonical sigma algebra. Therefore, the push forward of this stochastic random variable $W_n$ in the limit $n \to \infty$ can be defined as:

$$\bar{A}_j(s, \mathcal{X}) = \lim_{n \to \infty} \frac{1}{\sqrt{n}} \sum_{m=1}^{n} C_{j,m}^0 \varphi'(W_m^0 \cdot s) \sum_{i=1}^{d_s} s_i W_{d_s(m-1)+i},$$

which converges in distribution to a Normal distribution, with the mean and the variance are dependent on the state and the distribution $X_n$, by the Lindeberg–Lévy central limit theorem (Vershynin, 2018; Cai et al., 2019b) which is also a consequence of using GeLU activation which has bounded derivatives a.e. We drop the argument $X$ in the wherever it is implicit. The first term in the inequality, the norm of $\mu_j^n$, is upper-bounded as below:

$$\mathbb{E}\left[|\mu_j^n(s \wedge \tau^K) - \mu_j^n(t \wedge \tau^K)|^2\right] \lesssim_K \mathbb{E}\left[\left|\eta \sum_{\tau'=[s/\eta]\wedge\tau_K'/\eta}^{[t/\eta]\wedge\tau_K/\eta} \Phi_j^n G(W_{\tau'}^n)\right|^2\right]$$

$$\leq (t-s)^2 \|\Phi_j^n G(W_{\tau'}^n)\|_{L_\infty^n(E_K^n)}^2$$

$$\lesssim_{K,L_G,s} (t-s)^2,$$

where the second inequality is from the continuity of the function $\Phi_j^n G(W^n)$ in $W^n$. The last inequality is from the Lipschitz condition on the gradient function $G$ 5. For the second term, which is a martingale, in equation 17 we seek a similar bound to the one presented above:

$$\mathbb{E}\left[|\sigma_j^n(s \wedge \tau^K) - \sigma_j^n(t \wedge \tau^K)|^4\right] = \mathbb{E}\left[\left(\eta \sum_{\tau'=[s/\eta]\wedge\tau_K/\eta}^{[t/\eta]\wedge\tau_K/\eta} (\varsigma_j^{\tau'} - \varsigma_j^{\tau'-1})\right)^4\right]$$

$$= \mathbb{E}\left[\left(\eta \sum_{\tau'=[s/\eta]\wedge\tau_K/\eta}^{[t/\eta]\wedge\tau_K/\eta} \Phi_j^n H_n(W, \mathbb{B}_{W_{\tau'}})\right)^4\right]$$

$$\lesssim \mathbb{E}\left[\left(\eta^2 \sum_{\tau'=[s/\eta]\wedge\tau_K/\eta}^{[t/\eta]\wedge\tau_K/\eta} \left(\Phi_j^n H_n(W, \mathbb{B}_{W_{\tau'}})\right)^2\right)^2\right],$$

where the last inequality is from the Burkholder's inequality. We further expand the last term in the inequality as follows:

$$\mathbb{E}\left[\left(\eta^2 \sum_{\tau'=[s/\eta]\wedge\tau_K/\eta}^{[t/\eta]\wedge\tau_K/\eta} \left(\Phi_j^n H_n(W, \mathbb{B}_{W_{\tau'}})\right)^2\right)^2\right] = \eta^4 \sum_{\tau',\tau''} \mathbb{E}\left[\left(\Phi_j^n H_n(W, \mathbb{B}_{W_{\tau'}})\right)^2 \left(\Phi_j^n H_n(W, \mathbb{B}_{W_{\tau''}})\right)^2\right]$$

$$\leq \left(\eta \sum_{\tau'} \left(\eta^2 \mathbb{E}\left[\left(\Phi_j^n H_n(W, \mathbb{B}_{W_{\tau'}})\right)^4\right]\right)^{1/2}\right)^2$$

$$\lesssim_{K,L_H} (t-s)^2,$$

where the inequality in the second line is from Cauchy-Schwarz and for the last inequality we use the fact that (assumption 4) and the fact that $\eta \to 0$ at rate $O(\frac{1}{\sqrt{n}})$. This proves that $\sigma^n$ is 1/4 Hölder-continuous by Kolmogorov's continuity theorem. Since both the sequences $\mu_j^n$ and $\sigma_j^n$ are uniformly 1/2 Hölder-continuous we have that $v_j^n(s \wedge \tau_K)$ (equation 17) is also 1/2 Hölder-continuous. Further, we note that $v_j^n(s \wedge \tau_K)$ forms a tight sequence in $n$ with 1/2 Hölder-continuous limit point and $v_j^n(s \wedge \tau_K) - \mu_j^n(s \wedge \tau_K)$ is a martingale with a martingale limit point that is 1/4 Hölder-continuous limit point.

Now that we have proved that limit points exists and are 1/2 Hölder-continuous we seek to derive this limit. To do so we derive the quadratic variation

$$\sigma_j^n(t \wedge \tau^K)^2 - \int_0^t \eta \mathbb{E}_{\mathbb{B}_{W^{l/\eta}\wedge\tau_K}}\left[\left(\Phi_j^n H_n(W_{[l/\eta]\wedge\tau_K}, \mathbb{B}_{W_\tau})\right)^2\right],$$

which is a Martingale process. We seek to derive expression for the expectation above in the limit $n \to \infty$. To do so we derive the following:

$$\mathbb{E}_{\mathbb{B}_{W^{l/\eta}\wedge\tau_K}}\left[\left(\Phi_j^n H_n(W_{[l/\eta]\wedge\tau_K}, \mathbb{B}_{W_\tau})\right)^2\right] = \Phi_j^n \Xi_n(W_{[l/\eta]\wedge\tau_K})(\Phi_j^n)^\intercal,$$

where we omit the subscript $\mathbb{B}_{W_{l/\eta \wedge \tau_K}}$ under the expectation in the expression on right side for brevity. Letting $Y = W_{[l/\eta] \wedge \tau_K}$ and writing out rhe above expression based on equation 13

$$\Phi_j^n \Xi_n(Y)(\Phi_j^n)^\intercal = \frac{2}{3n^2} \mathbb{E}_{\mathbb{B}_Y \mathbb{B}'_Y} \left[ \sum_{m=1}^{n} \sum_{k=1}^{d_s} \sum_{m'=1}^{n} \sum_{k'=1}^{d_s} \varphi'(W_{m'}^0 \cdot s) s_{k'} \varphi'(W_m^0 \cdot s) s_k \left( \right. \right.$$

$$(C_{j,m'}^0)^4 \frac{1}{B^2} \sum_{b=1}^{B} \sum_{b'=1}^{B} q_j(s^b) \varphi'(W_m^0 \cdot s^b) s_k^b q_j(s^{b'}) \varphi'(W_{m'}^0 \cdot s^{b'}) s_{k'}^{b'}$$

$$- (C_{j,m'}^0)^4 \frac{1}{B} \sum_{b=1}^{B} q_j(s^b) C \varphi'(W_m^0 \cdot s^b) s_k^b G'_{j,m'(d_s-1)+k'}(Y)$$

$$- (C_{j,m'}^0)^4 \frac{1}{B} \sum_{b'=1}^{B} q_j(s^{b'}) \varphi'(W_{m'}^0 \cdot s^{b'}) s_k^{b'} G'_{j,m(d_s-1)+k}(Y)$$

$$\left. \left. + (C_{j,m'}^0)^4 G'_{j,m(d_s-1)+k}(Y)^2 \right) \right] + O\left(\frac{1}{n}\right).$$

Given that the gradient updates have finite and bounded variance (assumption 4) the expression including the expectation converges to a value that is $O(1)$ and dependent on $s, G(Y), j, \sigma_{H,K}$ by the strong law of large numbers at the rate $O\left(\frac{1}{n}\right)$. We have therefore have the following

$$\lim_{n \to \infty} \eta_n \Phi_j^n \Xi_n(W_{[l/\eta] \wedge \tau_K})(\Phi_j^n)^\intercal = 0.$$

Therefore, as $n \to \infty$ and by localization technique (Karatzas & Shreve, 2014) we prove convergence of equation 17 to:

$$d\bar{A}_j(s; W_t) = v(s; \mathcal{X}_t)dt, \tag{18}$$

which admits a unique solution due to the assumption of Lipschitz condition ,assumption 5. Given that we initialise $W_0$ as drawn from a distribution in $\mathbb{P}(\mathbb{R})$ then the distribution of $A_j$ is a push forward of this distribution and therefore evolves as in equation 18 and gives us the result. $\square$

Similarly, from the linearity of $A_{j,k}$ in $W$ using a similar derivation as above we can derive an ODE. To do so we first note

$$A_{j,k}^n(s; W) = \Phi_j^n(\mathbb{1}_k)W + \Phi_{j,k}^n(s)W,$$

where $\mathbb{1}_k$ is a $d_s$-dimensional vector with value at index $k$ is set to 1 and rest 0, $\Phi_{j,k}(s)$ is defined as below

$$\Phi_{j,k}^n(s) = \frac{1}{\sqrt{n}} \left[ W_{1,k}^0 C_{1,j}^0 \varphi''(W_1^0 \cdot s) s^\intercal, W_{2,k}^0 C_{2,j}^0 \varphi''(W_2^0 \cdot s) s^\intercal, ..., W_{n,k}^0 C_{n,j}^0 \varphi''(W_n^0 \cdot s) s^\intercal \right] \in \mathbb{R}^{1 \times nd_s}.$$

Therefore, in the limit $n \to \infty$

$$d\bar{A}_{j,k}(s) = v'(s; \mathcal{X}_t)dt.$$

Now we derive and prove the dynamics of the quadratic term $A_j(s; W)A_{j'}(s; W)$.

**Lemma 9.** *Given a fixed state $s$ the $j$-th action, $A_j^n(s; \cdot)$, determined by a linearised neural policy with two hidden layers as described and initialised in Section 3, we assume $W_0 \sim \mathcal{X}_0 = Normal(0, I_{d_s}/d_s)$ i.i.d whose gradient dynamics are described in equation 8 with learning rates $\eta_n \to 0$, and under assumptions 4, 5, we have that in the limit $n \to \infty$ the dynamics of $A_j^n$ converge weakly to the following random ODE*

$$d\bar{A}_j(s; \mathcal{X}_\tau)\bar{A}_{j'}(s; \mathcal{X}_\tau) = \left( v_j(s; \mathcal{X}_\tau)\bar{A}_{j'}(s; \mathcal{X}_\tau) + v_{j'}(s; \mathcal{X}_\tau)\bar{A}_j(s; \mathcal{X}_\tau) \right) d\tau, \tag{19}$$

*with the random variable $\mathcal{X}_\tau$ is the limit point of the sufficient random variables, $\mathcal{X}_t^n$, of the parameters updated according to stochastic policy gradient based updates laid out in Section 3.2 with $\eta = \eta_n$, and $v_j, v_{j'}$ are as described in Lemma 8.*

$$A_j^n(s; W) A_{j'}^n(s; W) = (\Phi_j^n(s; W_0) W)(\Phi_{j'}^n(s; W_0) W).$$

*Proof.* Since we know that $\bar{A}_j^\tau, \bar{A}_{j'}^\tau$ follow the ODE in equation 15 we want to show that the update for $A_j^n(s; W) A_{j'}^n(s; W)$ converges weakly to

$$d(\bar{A}_j(s; \mathcal{X}_t) \bar{A}_{j'}(s; \mathcal{X}_t)) = \big(v_j(s; \mathcal{X}_t) \bar{A}_{j'}(s; \mathcal{X}_t) + v_{j'}(s; \mathcal{X}_t) \bar{A}_j(s; \mathcal{X}_t)\big) dt$$

To do so consider the increments as in equation 16 for the statistic which a product of two actions $A_j^n A_{j'}^n$:

$$\nu_{j,j'}^\tau - \nu_{j,j'}^{\tau-1} = \Phi_j^n G_n(W_\tau)(\Phi_{j'}^n W_\tau) + (\Phi_j^n W_\tau) \Phi_{j'}^n G_n(W_\tau),$$
$$\varsigma_{j,j'}^\tau - \varsigma_{j,j'}^{\tau-1} = \big(\Phi_j^n H_n(W_\tau, \mathbb{B}_{W_\tau})\big) \Phi_{j'}^n W_\tau + \big(\Phi_j^n W_\tau\big) \Phi_{j'}^n H_n(W_\tau, \mathbb{B}_{W_\tau})$$
$$+ \eta \big(\big(\Phi_j^n H_n(W_\tau, \mathbb{B}_{W_\tau})\big) \Phi_{j'}^n G_n(W_\tau) + \big(\Phi_j^n G_n(W_\tau)\big) \Phi_{j'}^n H_n(W_\tau, \mathbb{B}_{W_\tau})\big)$$

Omitting the subscript in $\eta_n$ we obtain

$$u_j^\tau = u_j^0 + \eta \sum_{\tau'=1}^\tau (\nu_j^{\tau'} - \nu_j^{\tau'-1}) + \eta \sum_{\tau'=1}^{\tau'} (\varsigma_j^{\tau'} - \varsigma_j^{\tau'-1}).$$

Now for $l \in [0, L]$ we define,

$$\nu_j'(l) = \nu_j^{[l/\eta]} - \nu_j^{[l/\eta]-1}, \varsigma_j'(l) = \varsigma_j^{[l/\eta]} - \varsigma_j^{[l/\eta]-1},$$

Similar to the rpevious proof, we let

$$\mu_j^n(l) = \int_0^l \nu_j'(l) dl' = \nu_j'(\eta[l/\eta]) + (s - \eta[l/\eta]) \left(\nu_j^{[l/\eta]} - \nu_j^{[l/\eta]-1}\right)$$

$$\sigma_j^n(l) = \int_0^l \varsigma_j'(l) dl' = \varsigma_j'(\eta[l/\eta]) + (s - \eta[l/\eta]) \left(\varsigma_j^{[l/\eta]} - \varsigma_j^{[l/\eta]-1}\right),$$

be continuous linear interpolations based on discrete random variables $\nu, \varsigma$ and combine them together to obtain

$$v_j^n(l) = v_j^n(0) + \mu_j^n(l) + \sigma_j^n(l). \tag{20}$$

With exit time $\tau_K$ we want to show that for all $0 \le s, t \le T$,
$$\mathbb{E}||v_j^n(s \wedge \tau_K) - v_j^n(t \wedge \tau_K)||^4 \lesssim_{K,T} (t - s)^2,$$
where the expectation over the stochastic updates. This proves that $v_j^n(s \wedge \tau_K)$ is 1/4 Hölder-continuous according to Kolmogorov's continuity theorem (as opposed to 1/2 in the previous proof).

We have for all $s, t$
$$||v_j^n(s) - v_j^n(t)|| \le ||\mu_j^n(s) - \mu_j^n(t)|| + ||\sigma_j^n(s) - \sigma_j^n(t)||.$$

The first term in the inequality, the norm of $\mu_j^n$, is upper-bounded as below:

$$\mathbb{E}\left[|\mu_j^n(s \wedge \tau^K) - \mu_j^n(t \wedge \tau^K)|^4\right] \lesssim_K \mathbb{E}\left[\left|\eta \sum_{\tau'=[s/\eta] \wedge \tau_K'/\eta}^{[t/\eta] \wedge \tau_K/\eta} \Phi_j^n G_n(W_{\tau'})(\Phi_{j'}^n W_{\tau'})\right.\right.$$
$$\left.\left. + (\Phi_j^n W_{\tau'}) \Phi_{j'}^n G_n(W_{\tau'})\right|^4\right]$$
$$\le (t - s)^4 ||(\Phi_j^n W_{\tau'}) \Phi_{j'}^n G_n(W_{\tau'})$$
$$+ \Phi_j^n G_n(W_{\tau'})(\Phi_{j'}^n W_{\tau'})||_{L^\infty(E_K^n)}^4$$
$$\lesssim_{K,L_G,s} (t - s)^4.$$

where the second inequality is from the continuity of the function $\Phi_j^n W_{\tau'})\Phi_{j'}^n G_n(W_{\tau'})$ in $W^n$. The last inequality is from the Lipschitz condition on the gradient function $G$ 5.

Further consider the second term

$$
\mathbb{E}\left[|\sigma_j^n(s \wedge \tau^K) - \sigma_j^n(t \wedge \tau^K)|^4\right] = \mathbb{E}\left[\left(\eta \sum_{\tau'=[s/\eta]\wedge\tau_K/\eta}^{[t/\eta]\wedge\tau_K/\eta} (\varsigma_j^{\tau'} - \varsigma_j^{\tau'-1})\right)^4\right]
$$

$$
= \mathbb{E}\left[\left(\eta \sum_{\tau'=[s/\eta]\wedge\tau_K/\eta}^{[t/\eta]\wedge\tau_K/\eta} \varsigma_j^{\tau'} - \varsigma_j^{\tau'-1})\right)^4\right]
$$

$$
\lesssim \mathbb{E}\left[\left(\eta^2 \sum_{\tau'=[s/\eta]\wedge\tau_K/\eta}^{[t/\eta]\wedge\tau_K/\eta} \left(\varsigma_j^{\tau'} - \varsigma_j^{\tau'-1}\right)^2\right)^2\right].
$$

Using Cauchi-Schwarz inequality we can bound the two different types of terms separately. We can first upper bound

$$
\mathbb{E}\left[\left(\eta^2 \sum_{\tau'=[s/\eta]\wedge\tau_K/\eta}^{[t/\eta]\wedge\tau_K/\eta} \left(\Phi_j^n H_n(W_{\tau'})\right)\Phi_{j'}^n W_{\tau'}\right)^2\right]
$$

$$
= \eta^4 \sum_{\tau',\tau''} \mathbb{E}\left[\left(\left(\Phi_j^n H_n(W_{\tau'})\right)\Phi_{j'}^n W_{\tau'}\right)^2 \left(\left(\Phi_j^n H_n(W_{\tau''})\right)\Phi_{j'}^n W_{\tau''}\right)^2\right]
$$

$$
\le \left(\eta \sum_{\tau'} \left(\eta^2\mathbb{E}\left[\left(\left(\Phi_j^n H_n(W_{\tau'})\right)\Phi_{j'}^n W_{\tau'}\right)^4\right]\right)^{1/2}\right)^2
$$

$$
\lesssim_{K,L_H,s} (t-s)^2,
$$

where the inequality in the second line is from Cauchy-Schwarz and for the last inequality we use assumption 4, $\eta \to 0$ at rate $\frac{1}{\sqrt{n}}$ and the fact that $\Phi_j$ only depends on $s$. Similarly, we bound the other term below

$$
\mathbb{E}\left[\left(\eta^4 \sum_{\tau'=[s/\eta]\wedge\tau_K/\eta}^{[t/\eta]\wedge\tau_K/\eta} \left(\Phi_j^n H_n(W_{\tau'})\right)\Phi_{j'}^n G_n(W_{\tau'})\right)^2\right]
$$

$$
= \eta^8 \sum_{\tau',\tau''} \mathbb{E}\left[\left(\left(\Phi_j^n H_n(W_{\tau'})\right)\Phi_{j'}^n G_n(W_{\tau'})\right)^2 \left(\left(\Phi_j^n H_n(W_{\tau''})\right)\Phi_{j'}^n G_n(W_{\tau''})\right)^2\right]
$$

$$
\le \left(\eta^6 \sum_{\tau'} \left(\eta^2\mathbb{E}\left[\left(\left(\Phi_j^n H_n(W_{\tau'})\right)\Phi_{j'}^n G_n(W_{\tau'})\right)^4\right]\right)^{1/2}\right)^2
$$

$$
\lesssim_{K,L_G} (t-s)^2.
$$

Where the last inequality follows from Assumptions 4, convergence of $\eta \to 0$ and Lipschitz continuity of $G$. Combining these together we obtain the 1/2-Hölder-continuous limit point. Finally, to derive the dynamics we once again show that the quadratic variation goes to 0.

$$
\sigma_j^n(t \wedge \tau^K)^2 - \int_0^t \eta\mathbb{E}_{\mathbb{B}_{W^{l/\eta\wedge\tau_K}}}\left(\left(\Phi_j^n H_n(W_\tau)\right)\Phi_{j'}^n W_\tau + \left(\Phi_j^n W_\tau\right)\Phi_{j'}^n H_n(W_\tau)\right)^2.
$$

Since we have already shown that $\mathbb{E}_{\mathbb{B}_{W^{l/\eta\wedge\tau_K}}}\left[\left(\Phi_j^n H_n(W_{[l/\eta]\wedge\tau_K})\right)^2\right] = O(1) + O(1/n)$ (see Section C), similarly we have $(\Phi_{j'}^n W_\tau)^2 = O(1) + O(1/n)$. Therefore, as $n \to \infty$ and by localization technique we have as required:

$$
d(\bar{A}_j(s;\mathcal{X}_t)\bar{A}_{j'}(s;\mathcal{X}_t)) = \left(v_j(s;\mathcal{X}_t)\bar{A}_{j'}(s;\mathcal{X}_t) + v_{j'}(s;\mathcal{X}_t)\bar{A}_j(s;\mathcal{X}_t)\right)dt
$$

$\square$

Similarly, we can show the convergence to a random ODE for the product of two linear terms $A_{jk}A_{j'}$.

## G  PROOF OF MAIN RESULT

*Proof.* We put together the dynamics we have derived and proved convergence in the $n \to \infty$ limit. We re-arrange the terms of second order expansion of the Lie series to group together the terms that have the same multiplicative vector. We additionally denote the vector $\frac{\partial g(s)}{\partial s_k}$ by $g'_k(s)$ similarly $\frac{\partial h_j(s)}{\partial s_k}$ as $h'_{j,k}(s)$. From the derivation in section B we have the following:

$$
e_t^{X(W)}(s) = s + t \left( g(s) + \sum_{j=1}^{d_a} h_j(s) A_j(s; W) \right)
$$

$$
t^2 \left( \sum_{k=1}^{d_s} \left( g_k(s) g'_k(s) + \sum_{j=1}^{d_a} h_{j,k}(s) A_j(s, W) g'_k(s) \right. \right.
$$

$$
+ g_k(s) \left( \sum_{j'=1}^{d_a} h'_{j',k}(s) A_{j'}(s, W) + A_{j',k}(s, W) h_{j'}(s) \right)
$$

$$
\left. \left. + \sum_{j=1}^{d_a} \sum_{j'=1}^{d_a} h_{j,k}(s) A_j(s, W) h'_{j',k}(s) A_{j'}(s, W) + h_{j,k}(s) A_j(s, W) A_{j',k}(s) h_{j'}(s) \right) \right)
$$

$$
= s + \left( t g(s) + t^2 \sum_{k=1}^{d_s} g_k(s) g'_k(s) \right)
$$

$$
+ \sum_{j=1}^{d_a} t h_j(s) \left( A_j(s, W) + t \left( \sum_{k=1}^{d_s} g_k(s) \left( A_{j',k}(s, W) + \sum_{j'=1}^{d_a} h_{j',k}(s) A_{j'}(s, W) A_{j,k}(s) \right) \right) \right)
$$

$$
+ t^2 \sum_{j'=1}^{d_a} A_{j'}(s, W) \left( \sum_{k=1}^{d_s} h'_{j',k}(s) \left( g_k(s) + \sum_{j=1}^{d_a} h_{j,k}(s) A_j(s, W) \right) \right).
$$

The first four expressions in the summation account for $d_a + 2$ degrees of freedom. In other words, the first three terms in the summation are spanned by $d_a + 2$ vectors. For the last term in the summation consider the following representation:

$$
f_{j'}^\tau = A_{j'}^\tau(s) \left( \sum_{k=1}^{d_s} h'_{j',k}(s) \left( h_{j,k}(s) A_j^\tau(s) \right) \right)
$$

$$
= A_j^\tau(s) \sum_{k=1}^{d_s} h'_{j',k}(s) \sum_{j'=1}^{d_a} A_{j'}^\tau(s) h_{j',k}(s),
$$

$$
= A_j^\tau(s) J h_j(s) h(s) A^\tau(s),
$$

Where $J h_j$ is the Jacobian of the function $h_j(s)$. This leads us to the following vector:

$$
v_j^\tau = \sum_{k=1}^{d_s} \frac{\partial h_j(s)}{\partial s_k} \sum_{j'=1}^{d_a} \bar{a}_{j'}^\tau(s) h_{j',k}(s),
$$

$$
= J h_j(s) h(s) \bar{B}_j^\tau(s),
$$

$\bar{B}_j^\tau$ represents the $d_a \times 1$ vector $[\mathbb{E}[A_j^\tau A_1^\tau], \ldots, \mathbb{E}[A_j^\tau A_1^\tau]]$, where the expetation is over the stochasticity of initialisation..

where $J h_j(s)$ is the $d_s \times d_s$ Jacobian of $h_j$ w.r.t $s$, $\bar{a}_j^\tau(s)$ is the process determined by $\mathbb{E}\left[\nu_j^\tau\right]$, and $h(x)$ is a concatenation of $d_a$ vectors. We seek to upper bound the following, by showing that there

is a continuous time martingale process in the limit $n \to \infty$,

$$D(f^\tau, v^\tau) = D(\sum_{j=1}^{d_a} f_j^\tau, \mathrm{Span}(v_1^\tau, \ldots, v_{d_a}^\tau)),$$

where $D(f^\tau, v^\tau)$ is the distance between $f^\tau$ and the span of $v_1^\tau, \ldots, v_{d_a}^\tau$. This distance is upper bounded as follows:

$$D(f^\tau, v^\tau) \leq L^2 (\sum_{j=1}^{d_a} D(f_j^\tau, v_j^\tau))$$

$$\leq L^2 \left( \sum_{j=1}^{d_a} D(f_j^\tau, v_j^\tau) \right)$$

$$\leq L^2 \left( \sum_{j=1}^{d_a} a_j^\tau(s) J h_j(s) h(s) A^\tau(s) - J h_j(s) h(s) \bar{B}_j^\tau(s) \right)$$

$$\leq L^2 \left( \sum_{j=1}^{d_a} J h_j(s) h(s) (a_j^\tau(s) A^\tau(s) - \bar{B}_j^\tau) \right),$$

where both $\bar{a}_j, \bar{A}$ are the mean processes, and $L^2$ denotes the L-2 norm. Therefore, the data is concentrated around the space spanned by the vectors: $h_1, \ldots, h_{d_a}, v_1^\tau, \ldots, v_{d_a}^\tau$ and the paraboloid $tg + t^2 g'$. Let this product space be $M$ then $\dim(M) \leq 2d_a + 1$. The concentration property is a result of the concentration of $a_j^\tau(s) A^\tau(s)$ around $\bar{B}_j^\tau$ due to the dynamics in 19.

$\square$

While proofs in Appendix F closely follow that of Ben Arous et al. (2022) we list our contibtuions in this work:

1. We show that the distribution of outputs, and its quadratic combinations, of a two-layer linearised NNs deviate only in mean and variance, and are dependent on a finite set of summary statistics, despite the width and parameter size going to infinity as the learning rate goes 0.

2. In this appendix section, We combine this with the idea of the exponential map being a push forward of the parameter distribution at gradient time step $\tau$, for a fixed state $s$, and show that the distribution is concentrated around a low-dimensional manifold.

## H CONTINUOUS TIME POLICY GRADIENT

In this section we define the continuous time policy gradient. The policy gradient is similar to the one for discrete time introduced in Appendix J. We define the $i$-th component of the gradient over the value function with respect to the actions as follows:

$$(\nabla_a Q^\pi(s, a, t))_i = \lim_{h \to 0^+} \frac{Q_h^\pi(s, a + h e_i, t) - V^\pi(s, t)}{h},$$

where $e_i$ is a $d_a$ dimensional vector with 1 at $i$ and 0 otherwise, finally the function $Q_h$ is defined as:

$$Q_h^\pi(s, a + h e_i, t) = v^\pi(s_{t+h}) + \int_0^h e^{-\frac{l+t}{\lambda}} f_r(s_l^{a+he_i}) dl, \text{ such that}$$

$$s_l^{a+he_i} = \mathcal{T}(s, a + h e_i, l),$$

where $\mathcal{T}$ is the transition function as defined in Section 2.1. This is an application of the policy gradient theorem (Sutton et al., 1999; Lillicrap, 2015) which is a *semi-gradient* based optimisation technique.

It is not always guaranteed that the policy gradient algorithm will converge to globally optimal policies for general dynamics in a straightforward manner (Sutton et al., 1999; Konda & Tsitsiklis, 1999; Marbach & Tsitsiklis, 2003; Xiong et al., 2022). The rate of convergence affects the constant $\mathcal{R}_\tau$ introduced above. While we do not argue this formally, our main result holds as long as the RL algorithm does not diverge.

## I   DIMENSIONALITY ESTIMATION

We describe the algorithm for dimensionality estimation in the context of sampled data from the state manifold $\mathcal{S}_e$. Let the dataset be randomly sampled points from a manifold $\mathcal{S}_e$ embedded in $\mathbb{R}^{d_s}$ denoted by $\mathcal{D} = \{s_i\}_{i=1}^N$. For a point $s_i$ from the dataset $\mathcal{D}$ let $\{r_{i,1}, r_{i,2}, r_{i,3}, ...\}$ be a sorted list of distances of other points in the dataset from $s_i$ and they set $r_0 = 0$. Then the ratio of the two nearest neighbors is $\mu_i = r_{i,2}/r_{i,1}$ where $r_{i,1}$ is the distance to the nearest neighbor in $\mathcal{D}$ of $s_i$ and $r_{i,2}$ is the distance to the second nearest neighbor. Facco et al. (2017) show that the logarithm of the probability distribution function of the ratio of the distances to two nearest neighbors is distributed inversely proportional to the degree of the intrinsic dimension of the data and we follow their algorithm for estimating the intrinsic dimensionality. We describe the methodology provided by Facco et al. (2017) in context of data sampled by an RL agent from a manifold. Without loss of generality, we assume that $\{s_i\}_{i=1}^N$ are in the ascending order of $r_i$. We then fit a line going through the origin for $\{(\log(\mu_i), -\log(1-i/N)\}_{i=1}^N$. The slope of this line is then the empirical estimate of $\dim(\mathcal{S}_e)$. We refer the reader to the supplementary material provided by Facco et al. (2017) for the theoretical justification of this estimation technique. The step by step algorithm is restated below.

1. Compute $r_{i,1}$ and $r_{i,2}$ for all data points $i$.

2. Compute the ratio of the two nearest neighbors $\mu_i = r_{i,2}/r_{i,1}$.

3. Without loss of generality, given that all the points in the dataset are sorted in ascending order of $\mu_i$ the empirical measure of cdf is $i/N$.

4. We then get the dataset $\mathcal{D}_{\text{density}} = \{(\log(\mu_i), -\log(1-i/N)\}$ through which a straight line passing through the origin is fit.

The slope of the line fitted as above is then the estimate of the dimensionality of the manifold.

## J   DDPG BACKGROUND

An agent trained with the DDPG algorithm learns in the discrete time but with continuous states and actions. With abuse of notation, a discrete time and continuous state and action MDP is defined by the tuple $\mathcal{M} = (\mathcal{S}, \mathcal{A}, P, f_r, s_0, \lambda)$, where $\mathcal{S}, \mathcal{A}, s_0$ and $f_r$ are the state space, action space, start state and reward function as above. The transition function $P : \mathcal{S} \times \mathcal{A} \times \mathcal{S}$ is the transition probability function, such that $P(s, a, s') = \Pr(S_{t+1} = s'|S_t = s, A_t = a)$, is the probability of the agent transitioning from $s$ to $s'$ upon the application of action $a$ for unit time. The policy, in this setting, is stochastic, meaning it defines a probility distribution over the set of actions such that $\pi(s, a) = \Pr(A_t = a|S_t = s)$. The discount factor is also discrete in this setting such that an analogous state value function is defined as

$$v^\pi(s_t) = \mathbb{E}_{s_l, a_l \sim \pi, P}\left[\sum_{l=t}^T \lambda^{l-t} f_r(s_l, a_l)|s_t\right],$$

which is the expected discounted return given that the agent takes action according to the policy $\pi$, transitions according to the discrete dynamics $P$ and $s_t$ is the state the agent is at time $t$. Note that this is a discrete version of the value function defined in Equation 2. The objective then is to maximise $J(\pi) = v^\pi(s_0)$. One abstraction central to learning in this setting is that of the *state-action value function* $Q^\pi : \mathcal{S} \times \mathcal{A} \to \mathbb{R}$, for a policy $\pi$, is defined by:

$$Q^\pi = \mathbb{E}_{s_l, a_l \sim \pi, P}\left[\sum_{l=t}^T \lambda^{l-t} f_r(s_l, a_l)|s_t, a_t\right],$$

which is the expected discounted return given that the agent takes action $a_t$ at state $s_t$ and then follows policy $\pi$ for its decision making. An agent, trained using the DDPG algorithm, parametrises the policy and value functions with two deep neural networks. The policy, $\pi : \mathcal{S} \to \mathcal{A}$, is parameterised by a DNN with parameters $\theta^\pi$ and the action value function, $q : \mathcal{S} \times \mathcal{A} \to \mathbb{R}$, is also parameterised by a DNN with ReLU activation with parameters $\theta^Q$. Although, the policy has an additive noise, modeled by an Ornstein-Uhlenbeck process (Uhlenbeck & Ornstein, 1930), for exploration thereby making it stochastic. Lillicrap et al. (2016) optimise the parameters of the $Q$ function, $\theta^Q$, by optimizing for the loss

$$L_Q = \frac{1}{N} \sum_{i=1}^{N} (y_i - Q(s_i, a_i; \theta^Q))^2, \tag{21}$$

where $y_i$ is the target value set as $y_i = r_i + \lambda Q(s'_{i+1}, \pi(s_{i+1}; \theta^\pi); \theta^Q)$. The algorithm updates the parameters $\theta^Q$ by $\theta^Q \leftarrow \theta^Q + \alpha_Q \nabla_{\theta^Q} L_Q$, where $L_Q$ is defined as in Equation 21. The gradient of the policy parameters is defined as

$$\nabla_{\theta^\pi} J(\theta^\pi) = \frac{1}{N} \sum_i \nabla_a Q(s, a; \theta^Q)|_{s=s_i, a=\pi(s_i)} \nabla_{\theta^Q} \pi(s; \theta^\pi)|_{s=s_i}, \tag{22}$$

and the parameters $\theta^\pi$ are updated in the direction of increasing this objective.

## K  BACKGROUND ON SOFT ACTOR CRITIC

The goal of the SAC algorithm is to train an RL agent acting in the continuous state and action but discrete time MDP $\mathcal{M} = (\mathcal{S}, \mathcal{A}, P, f_r, s_0, \lambda)$, which is as described in Appendix J. The SAC agent optimises for maximising the modified objective:

$$J(\theta^\pi) = \sum_{t=0}^{T} \mathbb{E}_{s_t, a_t \sim \pi, P} \left[ f_r(s_t, a_t) + \mathcal{H}(\pi(\cdot, s_t; \theta^\pi)) \right],$$

where $\mathcal{H}$ is the entropy of the policy $\pi$. This additional entropy term improves exploration (Schulman et al., 2017; Haarnoja et al., 2017). Haarnoja et al. (2018) optimise this objective by learning 4 DNNs: the (soft) state value function $V(s; \theta^V)$, two instances of the (soft) state-action value function: $Q(s_1, a_t; \theta^Q_i)$ where $i \in \{1, 2\}$, and a tractable policy $\pi(s_t, a_t; \theta^\pi)$. To do so they maintain a dataset $\mathcal{D}$ os state-action-reward-state tuples: $\mathcal{D} = \{(s_i, a_i, r_i, s'_i)\}$. The soft value function is trained to minimize the following squared residual error,

$$J_V(\theta^V) = \mathbb{E}_{s \sim \mathcal{D}} \left[ \frac{1}{2} \left( V(s; \theta^V) - \mathbb{E}_{a \sim \pi} \left[ Q(s, a; \theta^Q) - \log \pi(s, a; \theta^\pi) \right] \right)^2 \right], \tag{23}$$

where the minimum of the values from the two value functions $Q_i$ is taken to empirically estimate this expectation. The soft $Q$-function parameters can be trained to minimize the soft Bellman residual

$$J_Q(\theta^Q) = \mathbb{E}_{s, a, r, s' \sim \mathcal{D}} \left[ \frac{1}{2} \left( Q(s, a; \theta^Q) - r - \lambda V(s'; \bar{\theta}^V) \right)^2 \right], \tag{24}$$

where $\bar{\theta}^V$ are the parameters of the target value function. The policy parameters are learned by minimizing the expected KL-divergence,

$$J(\theta^\pi) = \mathbb{E}_{s \sim \mathcal{D}} \left[ D_{KL} \left( \pi(s, \cdot; \theta^\pi), \frac{\exp(Q(s, \cdot; \theta^Q))}{Z_{\theta^Q}(s)} \right) \right], \tag{25}$$

where $Z_{\theta^Q}(s)$ normalizes the distribution.

## L  DDPG WITH GELU ACTIVATION

We provide the comparison between single hidden layer network and multiple hidden layer network because our results in Section 4 are for single hidden layer. The same architecture is used by Lillicrap et al. (2016) for the policy and value function DNNs which is two hidden layers of width 300

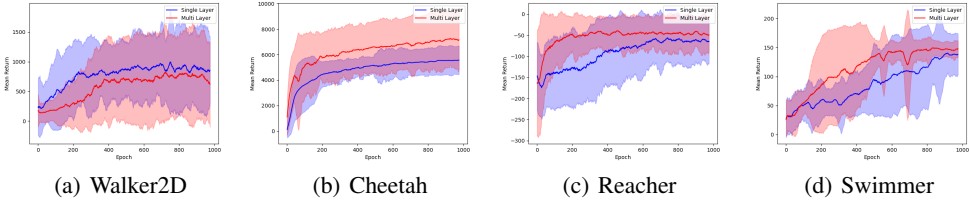

| (a) Walker2D | (b) Cheetah | (c) Reacher | (d) Swimmer |

Figure 6: Comparison of single hidden layer with GeLU activation (blue) and multiple hidden layer with ReLU activation (red) architectures for DNNs.

and 400 with ReLU activation. Here we provide the comparison to a single hidden layer width 400 with GeLU activation for the architecture used by Lillicrap et al. (2016). We provide this comparison in Figure 6 and note that the performance remains comparable for both architectures. All results are averaged over 6 different seeds. We use a PyTorch-based implementation for DDPG with modifications for the use of GeLU units. The base implementation of the DDPG algorithm can be found here:https://github.com/rail-berkeley/rlkit/blob/master/examples/ddpg.py. The hyperparameters are as in the base implementation.

## M  BACKGROUND ON SPARSE REPRESENTATION LEARNING VIA SPARSE RATE REDUCTION

We further explain and provide intuition on how the layer introduced in equation 12 learns a sparse representation. For a detailed explanation, we refer to the work by Yu et al. (2023a). While equation 12 performs a linear transform to the pre-activation of the layer, it is built upon the idea of *sparse rate reduction* that follows the principle of *parsimony*: learning representations by successively compressing and sparcifying the input signal to maximally differentiate different data points for the task (Wright & Ma, 2022).

We first establish notation in the setting of batched learning with neural networks. Suppose the objective is to represent $d$ data points as $n$-dimensional vectors. This problem can be formulated as obtaining a representation for $d \times n$, denoted by $\mathbf{Z}$ which represents a batch of data. The *coding rate*, as defined by Ma et al. (2007), is as follows,

$$R(\mathbf{Z}) = \frac{1}{2} \log \det \left( \mathbf{I} + \mathbf{Z}^\mathsf{T} \mathbf{Z} \right). \tag{26}$$

To promote sparsity, which in turn learns a low-rank representation of data with a non-linear low-dimensional structure, Yu et al. (2023a) optimize the following term:

$$\max_{\mathbf{Z}} \left[ R(\mathbf{Z}) - \lambda ||\mathbf{Z}||_0 \right] = \min_{\mathbf{Z}} \left[ \lambda ||\mathbf{Z}||_0 - \frac{1}{2} \log \det \left( \mathbf{I} + \frac{d}{n\epsilon^2} \mathbf{Z}^\mathsf{T} \mathbf{Z} \right) \right].$$

The iterative approach to learning a representation that minimizes the coding rate takes gradient steps in reducing this rate. As opposed to optimizing for these objectives in a loop, as in linear programming, Yu et al. (2023a) formulate this as representation learning over successive layers of NNs. Yu et al. (2023a) derive this iterative step, as representation over successive layers as follows,

$$\mathbf{Z}^{l+1} = \text{ReLU} \left( \left( 1 + \frac{4}{9(1 + \alpha')} \right) \mathbf{D}^\mathsf{T} \mathbf{Z}^l - \frac{4\lambda}{9\alpha'} \mathbf{I} \right),$$

where $\mathbf{Z}^l$ are assumed to be normalised, $\alpha'$ is the step size, and $\mathbf{D}^\mathsf{T}$ is assumed to be an orthogonal *dictionary* for the batched dataset. A parameterised version of this transform, where both the constant terms are interpreted as being independent, is used in equation 12. The sparsity layer adds a constant computation factor of $2n^2$ in the forward and backward passes, we show the impact on the steps per second metric in Figure 9 of the appendix.

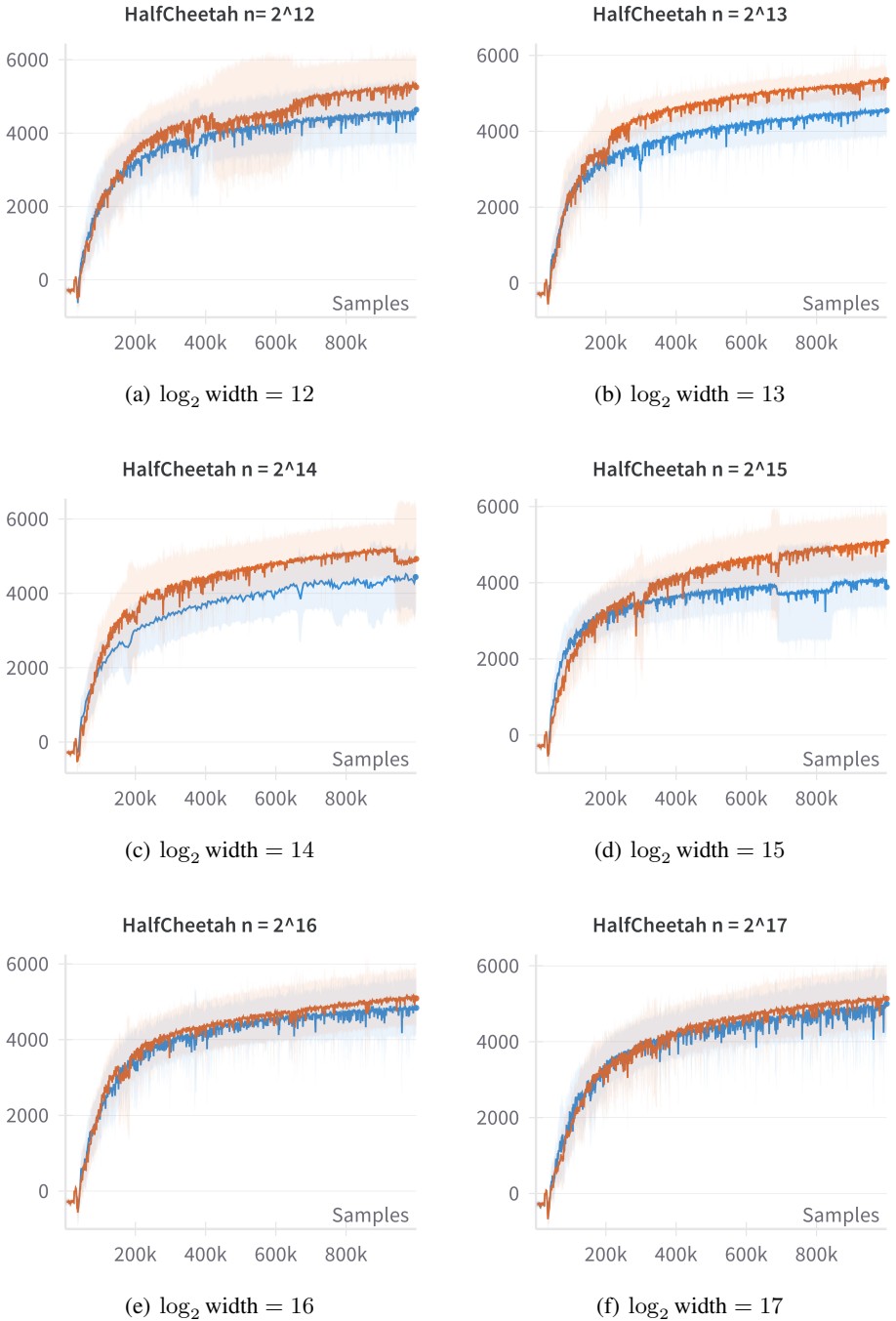

Figure 7: The canonical policy (in red) tracks the returns for linearised policy (in blue) at higher widths ($\log_2 n > 15$).

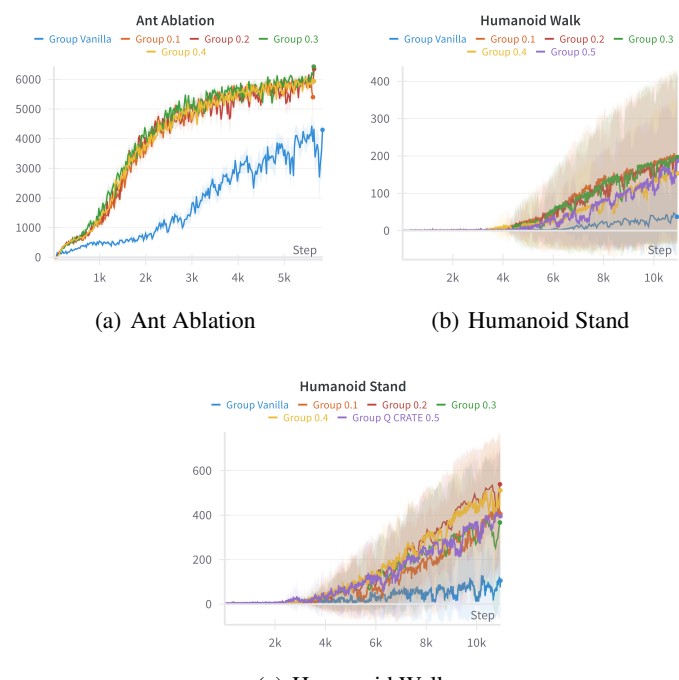

(a) Ant Ablation

(b) Humanoid Stand

(c) Humanoid Walk

Figure 8: Ablation over the $\alpha_Q$ parameter.

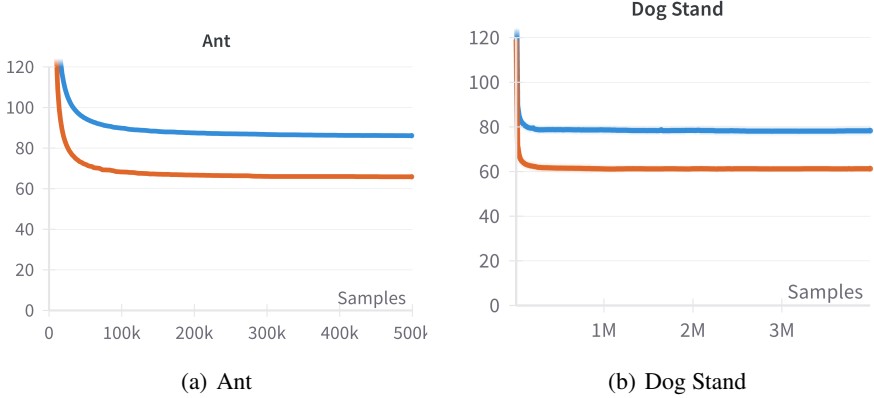

(a) Ant

(b) Dog Stand

Figure 9: We show the steps per second for SAC (blue) and sparse SAC (red) as training progresses. We observe that, in the ant environment, for the baseline the steps per second is as high as 90 whereas in the sparse implementation this drops to 65. Similarly, in the dog stand environment this difference is 81 to 61. While this increases the computation requirements and wall clock time it does not make our method intractable given the significant gains in performance.

## N    FURTHER EXPERIMENTAL RESULTS

We observe that the discounted returns don't vary for the ant MujoCo domain (Todorov et al., 2012) as shown in figure 8 with the environment steps on the x-axis. We see a lot of variance across and within choices of $\alpha_Q$ for humanoid walk and stand environment of DM control suite (Tunyasuvu-nakool et al., 2020) even though the sparse method remains superior to SAC with fully connected feedforward. We attribute this to being an exploration problem, while our method is able to overcome learning-related bottlenecks, it is unable to overcome the efficient exploration issue, which holds back the agent from attaining optimum returns in higher-dimensional control tasks.

## O    RELATED WORK

There has been significant empirical work that assumes the set of states to be a manifold in RL. The primary approach has been to study discrete state spaces as data lying on a graph which has an underlying manifold structure. Mahadevan & Maggioni (2007) provided the first such framework to utilise the manifold structure of the state space in order to learn value functions. Machado et al. (2017) and Jinnai et al. (2020) showed that PVFs can be used to implicitly define options and applied them to high dimensional discrete action MDPs (Atari games). Wu et al. (2019) provided an overview of varying geometric perspectives of the state space in RL and also show how the graph Laplacian is applied to learning in RL. Another line of work, that assumes the state space is a manifold, is focused on learning manifold embeddings or mappings. Several other methods apply manifold learning to learn a compressed representation in RL (Bush & Pineau, 2009; Antonova et al., 2020; Liu et al., 2021). Jenkins & Mataric (2004) extend the popular ISOMAP framework (Tenenbaum, 1997) to spatio-temporal data and they apply this extended framework to embed human motion data which has applications in robotic control. Bowling et al. (2005) demonstrate the efficacy of manifold learning for dimensionality reduction for a robot's position vectors given additional neighborhood information between data points sampled from robot trajectories. Continuous RL has been applied to continuous robotic control (Doya, 2000a; Deisenroth & Rasmussen, 2011; Duan et al., 2016) and portfolio selection (Wang et al., 2020; Wang & Zhou, 2020; Jia & Zhou, 2023; 2022b). We apply continuous state, action and time RL as a theoretical model in conjuction with a linearised model of NNs to study the geometry of popular continuous RL problem for the first time.

More recently, the intrinsic dimension of the data manifold and its geometry play an important role in determining the complexity of the learning problem (Shaham et al., 2015; Cloninger & Klock, 2020; Goldt et al., 2020; Paccolat et al., 2020; Buchanan et al., 2021; Tiwari & Konidaris, 2022) for deep learning. Schmidt-Hieber (2019) shows that, under assumptions over the function being approximated, the statistical risk deep ReLU networks approximating a function can be bounded by an exponential function of the manifold dimension. Basri & Jacobs (2017) theoretically and empirically show that SGD can learn isometric maps from high-dimensional ambient space down to $m$-dimensional representation, for data lying on an $m$-dimensional manifold, using a two-hidden layer neural network with ReLU activation where the second layer is only of width $m$. Similarly, Ji et al. (2022) show that the sample complexity of off-policy evaluation depends strongly on the intrinsic dimensionality of the manifold and weakly on the embedding dimension. Coupled with our result, these suggest that the complexity of RL problems and data efficiency would be influenced more by the dimensionality of the state manifold, which is upper bounded by $2d_a + 1$, as opposed to the ambient dimension.

We summarise several approaches for better representation learning in RL using information bottlenecks. Like our work, this approach reduces noise and irrelavant signal One common approach is to compress the state representation that is used by the agent for learning (Goyal et al., 2019a;b; 2020; Islam et al., 2022). The central idea is to extract the most informative bits with an auxiliary objective. This auxiliary objective could be exploration based (Goyal et al., 2019a), enables hierarchical decision making (Goyal et al., 2019b), predicting the goal (Goyal et al., 2020), and relevance to task dynamics (Islam et al., 2022). While these are practical methods they do not provide a theoretical limit on the dimension of the bottleneck. In contrast, our representation is a local manifold embedding that preserves the geometry of the emergent state manifold.

Another closely related line of research exploits the underlying structure and symmetries in MDPs. Ravindran & Barto (2001) provide a detailed and comprehensive study on on reducing the model size for MDPs by exploiting the redundancies and symmetries. There have been with other more specific approaches to this (Ravindran & Barto, 2003; 2002) and more recent work follow ups by van der Pol et al. (2020). The broader study of manifolds, within differential geometry, is related to the study of symmetries and invariances. We anticipate that further reducing the effective state manifold based on redundancies, to extend our work, would be highly promising. Givan et al. (2003) and Ferns et al. (2004) also provide closely related state aggregation techniques based on *bisimulation metrics* which have been developed further (Castro & Precup, 2010; Gelada et al., 2019; Zhang et al., 2020; Lan et al., 2021). The bi-simulation literature defines metrics that incorporates transition probabilities or environment dynamics of the environments. The underlying metric is probabilistic in nature. The manifold and metric are defined in such a way as to facilitate better representation learning for RL. The primary difference is that our approach proves how a low-dimensional manifold "emerges" from the design and structure of certain continuous RL problems.

We finally contextualize our work in light of various control theoretic frameworks. Control systems on a non-linear manifolds have been studied widely (Sussmann, 1973; Brockett, 1973; Nijmeijer & van der Schaft, 1990; Agrachev & Sachkov, 2004; Bloch & Bloch, 2015; Bullo & Lewis, 2019). Like most control theoretic frameworks the transitions, dynamics, and the geometry of the system are assumed known to the engineer. (Liu et al., 2021) recently provide a framework for controlling a robot on the *constraint manifold* using RL. As noted previously, our work is also closely related to the notion reachability in control theory (Kalman, 1960; Jurdjevic, 1997; Touchette & Lloyd, 1999; 2001) which deals with sets reachable under fixed and known dynamics of a system. Reachability sets from control theory have been applied for safe control under the RL framework (Akametalu et al., 2014; Shao et al., 2020; Isele et al., 2018). While the objective is similar, to find the sets of states reached with any controller, the assumptions, on the underlying dynamics are different leading to different results.

## P    EXTENSION TO OTHER ACTIVATIONS AND ARCHITECTURES

It is a difficult to theoretically analyse complex engineered systems such as neural networks for continuous control learned using policy gradient methods. We have simplified this setting by using linearised policies (Section 3.1) with GeLu activation and access to the true gradient of the value function (Section 2.1). We show results in GeLu activation because it is the closest (smooth) analogue to the most popularly used ReLu activation which is very commonly used in continuous control with RL (Lillicrap, 2015; Schulman et al., 2017; Haarnoja et al., 2018). Despite our choice of GeLu, as noted in Section 4, our results extend to activations which are twice differentiable everywhere with bounded derivatives. Moreover, our results capture the setting of neural policies that have a very high-dimensional parameter space but also have structured outputs (Lee et al., 2017; Ben Arous et al., 2022). In study of supervised deep learning results emanating from theoretical models that approximate shallow wide NNs have been extended to deeper networks, e.g. the neural tangeent kernel (NTK) framework (Jacot et al., 2018). Moreover, there have also been mechanisms to make finite depth and width corrections to NTK (Hanin & Nica, 2019). Theoretical inferences made in simplified settings have been extended to applications and a wide range of architectures as well (Yang & Hu, 2021; Yang et al., 2022; Fort et al., 2020; Wang et al., 2022). We anticipate that extending our results to a broader set of activations, architectures, and reinforcement learning algorithms would lead to better applications by means of improved theoretical understanding.

Another assumption we make is deterministic transitions. While this is true in many popular benchmark environments (Todorov et al., 2012; Tunyasuvunakool et al., 2020), the most general setting of RL as a model for intelligent agent the transitions are stochastic. This is a common feature in control theory where results in deterministic control: $\dot{s}(t) = g(s(t), u(t), t)$, with continuous states, actions, and time, can be extended to stochastic transitions by considering bounded stochastic perturbations

$$\dot{s}(t) = g(s(t), u(t), t) + d(s(t), u(t), t)dw_t,$$

where $d$ is the stochastic perturbation aspect with $w_t$ being the Wiener process and $u(t)$ is the open loop control. Tor example, the contraction analysis by Lohmiller & Slotine (1998) in deterministic transitions is extended to stochastic perturbations by Pham et al. (2009). We anticipate that our analysis, under appropriate assumptions on the stochastic perturbations, has promise of extensions.

