# OpenReview forum: "Geometry of Neural Reinforcement Learning in Continuous State and Action Spaces"
_ICLR.cc/2025/Conference — ICLR 2025 Oral_

### Official Review · Reviewer_7BEr · 2024-10-28

**Soundness:** 2
**Presentation:** 2
**Contribution:** 3
**Rating:** 6
**Confidence:** 3

**Summary:**

This work shows that training dynamics of a two layer neural policy induce a low dimensional manifold of attainable states embedded in the high-dimensional state space.

**Strengths:**

theoretical understanding

**Weaknesses:**

1. The manuscript is difficult to follow. Section 2 introduces a large number of mathematical concepts, including manifolds, fields, and Lie algebras. However, the connections between these mathematical concepts are fragile. I did not see the main storyline and the author's contribution clearly before page 7.
2. The paper primarily considers two-layer neural networks with GeLU activation. However, the subsequent theoretical analysis depends on linear approximation. I cannot tell the difference between this and single-layer neural networks and the impact of approximation errors on the proof results. Besides, extending the analysis to other network architectures and activation functions would strengthen the claims of universality.

**Questions:**

1. What does $\mathbb{B}_{W_{k \eta}}$ below equation (9) represent? Why does batch data need to come from SDE? Which research works on continuous time policy gradients have used this concept?
2. Does $G(W)$ in equation (10) contain $\eta$?
3. Since similar proof techniques were used, what are the innovative points of this manuscript compared to Ben Arous et al. (2022)?
4. The paper studied MDP for continuous-time systems. However, the simulation case is MuJoCo for discrete-time systems.
5. The gradient of policy network parameters for DDPG and SAC algorithms in simulation work deviates from the continuous time policy gradient. To what extent can simulation work verify the effectiveness of theoretical results?

---

> ### Author Response · Authors · 2024-11-22
> **Response to Reviewer 7BEr Part 1**
>
> Thank you for your time studying our manuscript and giving your valuable feedback. We address each concern individually below.
>
> > The manuscript is difficult to follow. Section 2 introduces a large number of mathematical concepts, including manifolds, fields, and Lie algebras. However, the connections between these mathematical concepts are fragile. I did not see the main storyline and the author's contribution clearly before page 7.
>
> This issue was also pointed out by other reviewers. We have whittled down the contents of section 2 to focus on the relevant details by moving some text to the appendix. Moreover, we have added explanations on how each one of these concepts relate to RL. Our main result section now starts on page 6 and we also provide context on how these results relate to the mathematical concepts explained in the background sections. We also note that despite our main result in section 4 we also introduce our learning setup in section 3. Please let us know if it addresses your concerns.
>
> > The paper primarily considers two-layer neural networks with GeLU activation. However, the subsequent theoretical analysis depends on linear approximation. I cannot tell the difference between this and single-layer neural networks and the impact of approximation errors on the proof results
>
> As pointed out by you, we show results in GeLu activation because it is the closest (smooth) analogue to the most popularly used ReLu activation. The linearised model we use is applied very commonly as a tractable approximation of neural networks (NNs) [1,2,3]. We refer to two layer neural networks which are the same as single hidden layer neural networks. Is that what you mean by a single layer NN? These are different from single layer NNs because the output is linear in NN parameters but not in the input, whereas a single layer linear NN is linear in the input. Learning in the wide linearised model of a two layer NN can be interpreted as the agent “picking and choosing” from a large number of randomly projected “features” to fit to the task, which in this case is an optimal policy. Moreover, Jaehoon et al [1] show that for supervised learning in the limit of width going to $\infty$ both the outputs and loss converge for training (Theorem H.1 and Section 2.4 “Infinite width networks are linearized networks” from their paper) and they back it up empirically (Figures S8, S9).
>
> We have now added an empirical comparison of discounted returns, for the Cheetah environment learned using a DDPG agent, between the linearised and canonical two layer NNs for increasing width in Section 5 (Figure 2) and report discounted returns as width increases in Figure 7 in the appendix . We observe that both the linearised and canonical neural networks have the same returns, across episodes, at very large width. Intuitively, this means that a linearised NN behaves similar to a canonical two layer NN when optimizing returns using a policy gradient algorithm. We are working on a theoretical result on how this approximation impacts our proof, on the estimated dimensionality, and we will post it as soon as we have it. Please let us know if this sufficiently addresses your concerns?
>
> > Besides, extending the analysis to other network architectures and activation functions would strengthen the claims of universality
>
> We note in section 4, that we require the following restrictions on the activation function:
>
> >>Furthermore, we assume that the activation, $\varphi$, has bounded first and second derivatives everywhere in $\mathbb R$.
>
> Our results apply to any activation which satisfies this condition (e.g. tanh). We have now added this explainer in the Appendix under the section: Extension to Other Architectures and Inputs. We also note how our results can lead to similar results for deeper fully connected networks. Our current work opens up room for obtaining theoretical results that apply to new architectures. Moreover, deep learning theory results in wide fully connected neural networks have been applied to obtain better theoretical understanding and applications of other neural network architectures [4,5,6,7,8]. We hope our results would further similar understanding of deep reinforcement learning which we now explain and note in the aforementioned Appendix section. As you might have noticed, our current manuscript is sufficiently explanatory of empirical settings and complex as is and extending it to more architectures might seem like a bit of an over stretch.

---

> > ### Author Response · Authors · 2024-11-22
> > **Response to Reviewer 7BEr Part 2 (answer to questions)**
> >
> > **We answer your questions below.**
> >
> > > 1. What does $\mathbb{B}{W{k \eta}}$ below equation (9) represent? Why does batch data need to come from SDE? Which research works on continuous time policy gradients have used this concept?
> >
> > $\mathbb{B}{W{k \eta}}$ represents the batch of data sampled at uniform random from the SDE, we have added this explanation. The expectation is over the different batches sampled in such a manner. In the past similar models have used this form of policy gradient in presence of varying levels of information and sampling techniques [10, 11] and also in continuous time actor-critic settings where the model is known [12].
> >
> > > 2. Does  $G(W)$ in equation (10) contain $\eta$?
> >
> > Yes it contains the term $k\eta$ and is also dependent on $\eta$, we are sorry about the oversight. We have now fixed this typo and also made section 3 more concise.
> >
> > > 3. ince similar proof techniques were used, what are the innovative points of this manuscript compared to Ben Arous et al. (2022)?
> >
> > In summary, we use the technique presented by Ben Arous et al. (2022) to derive the continuous time stochastic gradient ascent for policy gradient setting with the learning rate approaching zero. They “predict” that for finite dimensional inputs the dynamics of certain well behaved statistics will obey a random ordinary differential equation. In our work, we derive the dynamics of statistics which are either linear or quadratic in the parameters and also follow a random ODE. More appropriately, we show this for a “well behaved” (see assumptions in section 4) for a fixed state $s$ this is indeed a  random ODE. Our contribution in this part of the proof (for deriving this random ODE) is two fold:
> >
> > * We show that the distribution  of outputs, and its quadratic combinations, of a two layer linearised NN deviate only in mean and variance, and are dependent on a finite set of summary statistics, despite the width and parameter size going to infinity as the learning rate goes 0.
> >
> > * We combine this with the idea of the exponential map being a push forward of the parameter distribution at gradient time step $\tau$, for a fixed state $s$, and show that the distribution is concentrated around a low-dimensional manifold.
> >
> > We have now included this explanation in appendix E.  thanks to your review,  as a part of our proof sketch with insights into the bases of this low-dimensional manifold.
> >
> > > 4. The paper studied MDP for continuous-time systems. However, the simulation case is MuJoCo for discrete-time systems.
> >
> > Our goal, with a continuous time theoretical model, is to apply theory results to practice this despite the fact that most empirical research in RL for continuous states and actions is in discrete time. We will not find appropriate applications of continuous time results directly (barring a few exceptions). Moreover, for complex MuJoCo environments continuous time is impossible to simulate unless the dynamics are solved; these can be at best approximated (e.g. the Runge-Kutta method). Having said that, in section 5.2 with the toy problem we try to come as close as we can to continuous time setting by choosing $\delta t = 0.01$. This ensures that the discrepancy from a continuous time rollout is miniscule while demonstrating our upper bound. Thank you for pointing out that we have not made explicit this gap between theory and practice
> >
> > > 5. The gradient of policy network parameters for DDPG and SAC algorithms in simulation work deviates from the continuous time policy gradient. To what extent can simulation work verify the effectiveness of theoretical results?
> >
> > While DDPG comes closest to our theoretical setting, this question remains largely unanswered in our current work. We make a note of this in our discussion section. The work by Jia and Zhou [11] comes close to answering this, for a slightly different exploratory setting. We assume access to the true value function which is not always practical in RL. When theoretically analyzing complex and heavily engineered systems researchers have often had to make simplifying assumptions. Despite this, we are able to make significant improvements to SAC with a fairly straightforward change to the neural network.

---

> > > ### Author Response · Authors · 2024-11-22
> > > **Response to Reviewer 7BEr Part 3 (refrences)**
> > >
> > > **References:**
> > >
> > > [1] Wide neural networks of any depth evolve as linear models under gradient descent, Lee, Jaehoon and Xiao, Lechao and Schoenholz, Samuel and Bahri, Yasaman and Novak, Roman and Sohl-Dickstein, Jascha and Pennington, Jeffrey
> > >
> > > [2] Neural Policy Gradient Methods: Global Optimality and Rates of Convergence, Lingxiao Wang and Qi Cai and Zhuoran Yang and Zhaoran Wang
> > >
> > > [3] Neural temporal-difference learning converges to global optima, Qi Cai and Zhuoran Yang and Lee, {Jason D.} and Zhaoran Wang
> > >
> > > [4] Tensor Programs IV: Feature Learning in Infinite-Width Neural Networks, Yang, Greg and Hu, Edward J
> > >
> > > [5] Tensor Programs V: Tuning Large Neural Networks via Zero-Shot Hyperparameter Transfer, Greg Yang, Edward J. Hu, Igor Babuschkin, Szymon Sidor, Xiaodong Liu, David Farhi, Nick Ryder, Jakub Pachocki, Weizhu Chen, Jianfeng Gao
> > >
> > > [6] Deep learning versus kernel learning: an empirical study of loss landscape geometry and the time evolution of the neural tangent kernel, Stanislav Fort, Gintare Karolina Dziugaite, Mansheej Paul, Sepideh Kharaghani, Daniel M Roy, and Surya Ganguli.
> > >
> > > [7] When and why PINNs fail to train: A neural tangent kernel perspective, Wang, Sifan and Yu, Xinling and Perdikaris, Paris
> > >
> > > [8] Deep Learning Through A Telescoping Lens: A Simple Model Provides Empirical Insights On Grokking, Gradient Boosting \& Beyond, Alan Jeffares and Alicia Curth and Mihaela van der Schaar, 2024
> > >
> > > [9] Reinforcement Learning in Continuous Time and Space: A Stochastic Control Approach, Haoran Wang and Thaleia Zariphopoulou and Xun Yu Zhou
> > >
> > > [10] Policy gradient in continuous time, Remi Munos
> > >
> > > [11] Policy gradient and actor-critic learning in continuous time and space: Theory and algorithms, Jia, Yanwei and Zhou, Xun Yu
> > >
> > > [12] Reinforcement Learning In Continuous Time and , Kenji Doya

---

> > > > ### Comment · Reviewer_7BEr · 2024-11-25
> > > >
> > > > The response has solved most of my problems, and I have already modified my rate.

---

> ### Comment · Area_Chair_fGY4 · 2024-11-23
> **From AC.**
>
> Reviewer 7BEr: if possible, can you respond to the rebuttal?

---

### Official Review · Reviewer_S4jN · 2024-10-30

**Soundness:** 3
**Presentation:** 3
**Contribution:** 3
**Rating:** 8
**Confidence:** 3

**Summary:**

The paper proposes a theoretical approach to understanding reinforcement learning in continuous state and action environments using a geometric perspective. Unlike traditional theoretical RL models that focus on finite state and action spaces, this work introduces the concept of a low-dimensional manifold that captures the space of states achievable by RL agents trained with actor-critic methods. By demonstrating that the dimensionality of this manifold is bounded by the dimensionality of the action space, the authors establish that RL agents operate within a constrained subset of the full state space.

**Strengths:**

By proving that RL agents operate within a low-dimensional manifold of attainable states, the authors offer a mathematically rigorous insight that connects the geometry of state spaces to the action dimensionality.
This theoretical framework is supported by empirical evidence from simulated environments, such as MuJoCo, showing that training dynamics in RL indeed produce a low-dimensional representation. The paper presents a practical application by incorporating a manifold-learning layer in policy and value networks, which improves performance in complex control tasks.

**Weaknesses:**

The analysis assumes deterministic transitions and access to an exact value function, which is often impractical in dynamic, stochastic environments.
The simulation setups may not fully capture the complexity and variability of real-world tasks where environmental noise and high-dimensional data structures can complicate learning dynamics.
Lastly, the mathematical framework is limited to two-layer neural networks, which may oversimplify the behaviors of deeper architectures commonly used in modern RL, potentially limiting the generalizability of the findings to more complex neural network models.

**Questions:**

How could the theoretical framework be adapted to account for stochastic transitions and noisy reward functions, which are common in real-world environments?
Can the manifold-learning approach be tested with deeper and more complex neural network architectures to better reflect the structure of contemporary RL models?

---

> ### Author Response · Authors · 2024-11-22
> **Response to reviewer S4jN**
>
> Thank you for carefully going through our work and your supportive comments. We have tried to address in a lot more detail the primary weakness pointed out by:
>
> > The simulation setups may not fully capture the complexity and variability of real-world tasks …
>
> We have added a section in the appendix titled “Extension to Other Activations and Architectures”  on how applicable our results are to more complex neural network architectures and stochastic transitions. In summary, we remark how in the past theoretical analysis of shallow large width networks [1] have been extended to finite depths and widths [2]. We also remark how continuous control results from deterministic dynamics [3] have been extended to stochastic dynamics [4]. We make these arguments in much more detail with various examples.
>
> We recognise that this is a valid limitation of our work but there is precedent that simplified theoretical analyses leads to broader application.
>
> **We answer your questions below.**
>
> > How could the theoretical framework be adapted to account for stochastic transitions and noisy reward functions, which are common in real-world environments?
>
> Many popular RL benchmarks for continuous control have deterministic transitions (e.g. MuJoCo and DM Control). We agree that the more real world cases will be stochastic. Presenting theoretical results with deterministic transitions and extending them to stochastic transitions is more common. One such model is to provide results for the deterministic case $\dot{s}(t) = g(s(t), u(t), t)$ and then extend to the case with stochastic perturbation:
> $$\dot{s}(t) = g(s(t), u(t), t) +d(s(t), u(t), t) dw_t,$$
> where $d$ is the stochastic perturbation aspect with $w_t$ being the Wiener process and $u(t)$ is the open loop control. We now make this remark in the appendix.
>
> > Can the manifold-learning approach be tested with deeper and more complex neural network architectures to better reflect the structure of contemporary RL models?
>
> Many popular continuous control RL frameworks with sensory inputs utilise three layer NNs, which is what we benchmark against in action 5.3. At the same time contemporary methods employ latent transition models (e.g. Dreamer v3 [5]) which are learned using complex deep NNs. One possible application of our work could be to incorporate the low-dimensional aspects of the attainable set of states into these transition models. We also anticipate significant performance improvement in future work with deeper networks constructed using the sparsification layer we have used.
>
> --------
>
> **References:**
>
> [1] Neural Tangent Kernel: Convergence and Generalization in Neural Networks, Arthur Jacot, Franck Gabriel, Clément Hongler
>
> [2] Finite Depth and Width Corrections to the Neural Tangent Kernel, Boris Hanin, Mihai Nica
>
> [3] On contraction analysis for non-linear systems. Winfried Lohmiller and Jean-Jacques E. Slotine
>
> [4] A contraction theory approach to stochastic incremental stability. Quang-Cuong Pham, Nicolas Tabareau, and Jean-Jacques Slotine
>
> [5] Mastering diverse domains through world models, Hafner, Danijar and Pasukonis, Jurgis and Ba, Jimmy and Lillicrap, Timothy

---

> > ### Comment · Reviewer_S4jN · 2024-11-24
> >
> > Thank you for your responses; they have addressed my concerns. Since my rating is already positive, I will leave it as it is.

---

> ### Comment · Area_Chair_fGY4 · 2024-11-23
> **From AC**
>
> Reviewer S4jN: if possible, can you respond to the rebuttal?

---

### Official Review · Reviewer_zo6o · 2024-11-03

**Soundness:** 3
**Presentation:** 3
**Contribution:** 3
**Rating:** 8
**Confidence:** 2

**Summary:**

The paper presents a novel theoretical and empirical analysis of reinforcement learning (RL) within continuous state and action spaces by employing a geometric framework. The authors propose that the set of attainable states in RL systems is constrained within a low-dimensional manifold, which is directly influenced by the dimensionality of the action space. Specifically, they examine the structure of locally attainable states using semi-gradient updates in actor-critic learning, aiming to identify and exploit low-dimensional representations for RL in high-dimensional environments. This approach is validated through experiments on MuJoCo environments, demonstrating the practical implications and performance gains of using low-dimensional manifold learning in RL settings with high degrees of freedom.

**Strengths:**

- Strong theoretical insights: The authors look at the geometry of RL dynamics in continuous-time MDP with continuous spaces in order to link the dimensionality of the attainable state manifold to the action space. Theorem 1 is the main contribution in this reward. The theorem formally shows that the dimension of this manifold is related to the dimensionality of the agent's action space rather than the full state space. Specifically, the dimension of the manifold is approximately $2 \times (action dimension) + 1$. I found this insight quite interesting. It looks like the theorem suggests that by understanding this low-dimensional structure, we can design more efficient policies or networks, as we don’t need to account for the full high-dimensional state space, as seen in results in section 5.3.

- Interesting empirical experiments on multiple MuJoCo environments that support the theoretical claims (see above comment). By measuring the dimensionality of the attainable state space, the results indicate consistency with the derived bounds, highlighting the empirical validity of the proposed approach.

**Weaknesses:**

- The mathematical presentation is too complex: The derivation and presentation of the theoretical framework are mathematically dense, which could limit accessibility for a broader audience. The notation and terminology, especially in the sections on Lie series and vector fields, may be difficult for readers less familiar with differential geometry. It would be great if there could be some better insights following each main step.

-  There is little discussion on scalability: While the results are validated empirically, the paper could benefit from further discussion on the computational complexity and scalability of the approach, particularly in scenarios with much larger state-action spaces or image input space.

**Questions:**

- Q1: Is the sparsification layer computationally efficient compared to fully connected layers, particularly as the number of states and actions grows?

- Q2: How does this work compare in performance with other state representation techniques, such as latent variable models in RL?

In overall, the paper is generally well-written, with each section following logically from the previous one. However, the introduction to differential geometry concepts (e.g., Lie derivatives, exponential maps) could be more concise, as this level of detail might obscure the main contributions. The plots provided for MuJoCo environments are helpful, but additional figures explaining the theoretical framework would improve clarity.


This submission is a strong contribution to theoretical RL research and provides significant insights into the geometric structure of attainable states in continuous environments. The empirical validation in MuJoCo environments further solidifies the practical value of the proposed method.

* Minor comments

- Figure 2: What is the red line?

---

> ### Author Response · Authors · 2024-11-21
> **Response to reviewr zo6o**
>
> Thank you for your valuable and insightful review and pointers on how to improve our work. We have significantly re-organised the background section and also provide better insights following our main theoretical result. More specifically, we intuit what the bases of this low-dimensional manifold are. We have also added a discussion on scalability and address it in detail below.
>
> > The mathematical presentation is too complex: The derivation and presentation of the theoretical framework are mathematically dense, which could limit accessibility for a broader audience. The notation and terminology, especially in the sections on Lie series and vector fields, may be difficult for readers less familiar with differential geometry. It would be great if there could be some better insights following each main step.
>
> We apologise for the oversight in this aspect. We have now moved text that wasn't absolutely needed to understand our results and the main story to the appendix. We have also additionally provided intuitive explanations in sections 2.2 and 2.3 and how they relate to RL. We have also added a broader more intuitive proof sketch. Please take a look and let us know if this addresses your concerns.
>
> > There is little discussion on scalability: While the results are validated empirically, the paper could benefit from further discussion on the computational complexity and scalability of the approach, particularly in scenarios with much larger state-action spaces or image input space.
>
> The sparsity layer (equation in section 5.4) adds a constant computation factor of $2n^2$ in the forward and backward passes where $n$ is the width of the layer, the big-O computational complexity remains the same though. This is reflected in the metric steps per second. In the most recent version we report this in the Appendix section titled “Further Experimental Results”. We observe that, in the ant environment, for the baseline the steps per second is as high as ~90 whereas in the sparse implementation this drops to ~65. Similarly, in the dog stand environment this difference is ~81 to ~61.  While this increases the computation requirements and wall clock time it does not make our method intractable given the significant gains in performance. We expect this metric to be impacted as the input size increases.
>
> Image inputs are significantly different, in our opinion. The theoretical setting is that of the input size going to $\infty$ with the width (see chapter 2 in textbook by Romain and Zhenyu [1] or [2,3]). This is so because in the very high dimensional setting the input size (e.g. 256 X 256) is comparable to large widths of neural networks. We assume the input size is orders of magnitude smaller, which is true of control/sensor inputs in MujoCo and deepmind control suite.We make a note of this distinction in our discussion section but as of now we do not delve into how the manifold dimensionality and subsequent representation learning will change in this setting.
>
> We have answered Q1 above. For Q2:
>
> > Q2: How does this work compare in performance with other state representation techniques, such as latent variable models in RL?
>
> Do you mean latent state models like dreamerV3 [5] and such? Dreamer does not perform well, in fact learning does not progress at all, for dog and humanoid environments of deep mind control suite (see Figure 5 in [4]). These are particularly “notorious” high-dimensional environments where we show great improvement. We also note that model based methods with significant added complexity do perform better whereas we have simply made the neural net wider and modified a layer while keeping the number of parameters the same as in the baseline. Please let us know if that is not what you meant by “latent space models” or if we have not addressed your concern.
>
> > Figure 2: What is the red line?
>
> This was an error, thank you for pointing it out. We drew a red line at $d_a + 1$ but it was supposed to be at $2 d_a + 1$. We have made that fix with an explanation in the latest version of our paper.
>
> ------
>
> **References:**
>
> [1] Random matrix methods for machine learning, Couillet, Romain and Liao, Zhenyu
>
> [2] Phase diagram of stochastic gradient descent in high-dimensional two-layer neural networks, Veiga, Rodrigo and Stephan, Ludovic and Loureiro, Bruno and Krzakala, Florent and Zdeborov{\'a}, Lenka
>
> [3] Dynamics of stochastic gradient descent for two-layer neural networks in the teacher-student, Goldt, Sebastian and Advani, Madhu and Saxe, Andrew M and Krzakala, Florent and Zdeborov{\'a}, Lenka
>
> [4] Td-mpc2: Scalable, robust world models for continuous control, Hansen, Nicklas and Su, Hao and Wang, Xiaolong
>
> [5] Mastering diverse domains through world models, Hafner, Danijar and Pasukonis, Jurgis and Ba, Jimmy and Lillicrap, Timothy

---

> ### Comment · Area_Chair_fGY4 · 2024-11-23
> **From AC.**
>
> Reviewer zo6o: if possible, can you reply to the rebuttal?

---

> > ### Comment · Reviewer_zo6o · 2024-11-25
> > **Thanks**
> >
> > Thanks the authors for the response. It has addressed my concerns. I updated my score.

---

### Official Review · Reviewer_aRZw · 2024-11-04

**Soundness:** 4
**Presentation:** 3
**Contribution:** 4
**Rating:** 8
**Confidence:** 2

**Summary:**

This paper studies the Manifold hypothesis for deterministic continuous state and action environments in RL. In particular they show that the set of reachable states for a certain class of analytically tractable neural network policies learned using stochastic policy gradient lies on a manifold with a dimension upper bounded by a linear function of the action space dimension. They demonstrate this upper bound in empirical experiments and show that using a sparse representation can help in learning in complex environments.

**Strengths:**

The paper is well-written and rigorous.
The analysis of the manifold hypothesis for RL using neural network policies is novel to my knowledge, and a significant contribution to the area.

**Weaknesses:**

I don’t have any major comments on the weaknesses of the paper, I feel that the authors adequately mention the limitations in Section 6.

Minor:
- There were a few awkward or incomplete sentences that I did not understand:
  - L263-265: “In theoretical frameworks ….”
  - Caption of Figure 3
  - L462: “A common of a fully …”
  - The paragraph under Equation (14), and specifically the sentence in L475-478.
- The fonts in Figures 2,3, and 4 were too small to read.

**Questions:**

- In Figure 2, I thought that the upper bound is supposed to be $2d_a + 1$ but the caption says that the green line is $d_s$ and the estimated dimensionality is below it. Also what is the red line in this figure?

- In Figure 4, what is “Group” in the legend referring to?

---

> ### Author Response · Authors · 2024-11-21
> **Response to Reviewer aRZw**
>
> Thank you very much for your support and approval of our work. We have fixed the typos and mistakes you have mentioned. Once again, your valuable comments are much appreciated. We have also fixed the paragraph in section 5.2 as below:
>
> >>We instead evaluate the intrinsic dimension of the locally attainable set under feedback policies within our theoretical framework. We do so for the set of states attained for small $t$ under the dynamics $\dot{s(t)} = A s(t) + B \Phi(x) W$, where $A, B$ are as in equation above. To achieve this, for fixed embedding dimension $d_s$ we obtain neural networks sampled uniformly randomly from the family of linearised neural networks as in definition above, with $r = 1.0, t \in  (0, 5), n = 1024$. Consequently we obtain 1000 policies with $\delta t = 0.01$, and therefore a sample of 500000 states to estimate the intrinsic dimension of the attained set of states using the algorithm by Facco et al, 2017. We vary the dimensionality of the state space, $d_s$, from 3 to 10 to observe how the intrinsic dimension of the attained set of states varies with the embedding dimension while keeping $d_a$ fixed at 1. The dimensionality of the attained set of states remains upper-bounded by  $2d_a + 1 = 3$ for this system (figure 3).
>
> We have increased the font size on the images as well.  Answer to your questions:
>
> Q1:
> We have fixed that error, the red line is now at $2 d_a + 1$ it was previously at $d_a + 1$ which was an oversight on our side. The green line still remains at $d_s$ to show the discrepancy between the state space dimensionality and the attainable set dimensionality.
>
> Q2:
> Group in the legend is residue from the wandb interface which we use for tracking our learning algorithms; we have now removed it. Thank you for pointing it out.

---

> > ### Comment · Reviewer_aRZw · 2024-11-25
> >
> > Thank you for addressing my comments. I am happy to keep my score.

---

> ### Comment · Area_Chair_fGY4 · 2024-11-23
> **From AC.**
>
> Reviewer aRZw: if possible, can you reply to the rebuttal?

---

### Meta-Review · Area_Chair_fGY4 · 2024-12-19

**Metareview:**

The paper proposes a new regulariser for actor-critic methods (like SAC) that work in continuous action spaces. The regulariser works by constraining the state to be in a low-dimensional manifold. The paper has both an impressive theoretical derivation (a proof that, under regularity assumptions, attainable states are indeed on the manifold) as well as empirical evidence showing that the approach works well in practice.

The most significant shortcomings is the difficulty most people in the ICLR community will have parsing the paper. However, this is unavoidable to an extent given the complexity of the math used to derive the result.

Because the strengths heavily outweigh the shortcomings, I strongly recommend acceptance.

I also have the following last-minute request for the camera-ready version: please provide a more detailed explanation of what $\lambda_1$
is in line 503 as well as more generally a more complete discussion of how that equation enforces sparsity. Please also explain if the width of sparsification layer is hard-coded for each task separately (it is possible they already include this information in the paper, but I couldn't find it).

**Additional Comments On Reviewer Discussion:**

All reviewers agreed this should be accepted, with varying degrees of enthusiasm. I am reasonably enthusiastic myself and recommend an oral presentation. I understand it may have to be downgraded to a spotlight if there isn't enough slots for an oral etc.

---

### Decision · Program_Chairs · 2025-01-22

Accept (Oral)